# CURVATURE-INFORMED SGD VIA GENERAL PURPOSE LIE-GROUP PRECONDITIONERS

## ABSTRACT

We present a novel approach to accelerate stochastic gradient descent (SGD) by utilizing curvature information obtained from Hessian-vector products or finite differences of parameters and gradients, similar to the BFGS algorithm. Our approach involves two preconditioners: a matrix-free preconditioner and a low-rank approximation preconditioner. We update both preconditioners online using a criterion that is robust to stochastic gradient noise and does not require line search or damping. To preserve the corresponding symmetry or invariance, our preconditioners are constrained to certain connected Lie groups. The Lie group's equivariance property simplifies the preconditioner fitting process, while its invariance property eliminates the need for damping, which is commonly required in second-order optimizers. As a result, the learning rate for parameter updating and the step size for preconditioner fitting are naturally normalized, and their default values work well in most scenarios. Our proposed approach offers a promising direction for improving the convergence of SGD with low computational overhead. We demonstrate that Preconditioned SGD (PSGD) outperforms SoTA on Vision, NLP, and RL tasks across multiple modern deep-learning architectures.

## 1 INTRODUCTION

Optimizing machine learning models with millions of free parameters presents a significant challenge. While conventional convex optimization algorithms such as Broyden-Fletcher-Goldfarb-Shanno (BFGS), its limited-memory version, L-BFGS, conjugate gradient (CG), and nonlinear versions like Hessian-free (HF) optimization (Martens and Sutskever, 2012) have succeeded in small-scale, convex mathematical optimization problems, they are rarely used for large-scale, stochastic optimization problems that arise in machine learning (ML). One of the main reasons is their reliance on the line search step. In many ML models, such as variational and reinforcement learning models, cost functions are defined as expectations and can only be evaluated through Monte Carlo (MC) sampling averages. This can result in large variances, making optimizers that rely on line search to ensure convergence problematic. Several recent extensions of these methods to deep learning, such as K-BFGS and KFAC, have foregone the line search step in favor of damping (Goldfarb et al., 2020; Martens and Grosse, 2015). However, this adds complexity by introducing extra hyperparameters.

Empirical results indicate that plain SGD is a highly efficient optimizer for most ML problems. However, for problems with a large eigenvalue spread, SGD may converge slowly once the solution is located in a basin of attraction. Adaptive optimizers such as RMSProp and Adam (Kingma and Ba, 2015) converge faster but have been shown to generalize worse on many problems (Wilson et al., 2017; Zhou et al., 2020a). Reducing the generalization gap between SGD and Adam remains an active topic of research (Zhuang et al., 2020). This work focuses on providing SGD with a good preconditioner to accelerate its convergence around the basin of attraction without undermining its generalization capacity. The curvature information for preconditioner fitting can be sampled from Hessian-vector products or finite differences of parameters and gradients, similar to the BFGS algorithm. However, constructing a preconditioner in a deterministic way, as in BFGS (Boyd and Vandenberghe, 2004; Goldfarb et al., 2020), may not be possible due to potential issues with line search and damping. Therefore, we adopt the more general and gradient-noise-robust preconditioner fitting criterion proposed in Li (2015) and fit the preconditioner online with another "gradient descent" algorithm. The key is to avoid making the preconditioner fitting problem more difficult and computationally expensive than the original parameter-learning problem.

In this paper, we propose using Lie groups as a tool for preconditioner fitting. The "gradient descent" on a Lie group is similar to common gradient descent in Euclidean space. It involves applying a series of small transforms via multiplication with $I + \mu G$, where $\mu$ is a small scalar and $G$ is the group generator (see D.1). The Lie group is a rich and convenient space to work in since moving a preconditioner around any point on the group behaves similarly to moving it around the identity element of the group, i.e., the identity matrix $I$. This is known as the Lie group equivariance property.

Recent curvature-aware optimization methods such as HessianFree, KFAC, AdaHessian (Yao et al., 2021), K-BFGS, Shampoo (Gupta et al., 2018) and GGT (Agarwal et al., 2018) have shown good empirical results in Deep Learning (Osawa et al., 2022). Yet, they require damping, line search, or regret and are thus susceptible to pitfalls that do not affect PSGD where gradient noises regularize both parameter and preconditioner updates.

In an empirical setting, PSGD simultaneously surpasses the corresponding SoTA optimizers across vision, natural language processing (NLP), and reinforcement learning (RL) tasks, and estabishes new SoTA for many networks and settings, *e.g.* ResNet, LSTMs (Hochreiter and Schmidhuber, 1997) and GPT-2 (Radford et al., 2019). We consider optimization problems involving MNIST (Le-Cun and Cortes, 2010), CIFAR-10 (Krizhevsky, 2009), Penn Treebank (Marcus et al., 1993), the complete works of Shakespeare, the Open Web Text (OWT) dataset (Gokaslan and Cohen, 2019), HalfCheetah and RoboWalker (Brockman et al., 2016). PSGD outperforms SoTA methods with negligible overhead compared to SGD across a wide range of optimization problems. PSGD provides practitioners with a powerful, stable, and efficient optimization tool that can significantly enhance the performance of deep learning models in various domains.

## 2 BACKGROUND

### 2.1 NOTATIONS

The objective is to minimize a loss function defined as an expectation, $\mathcal{L}(\theta) = E_z[\ell(\theta, z)]$, where $\theta \in \mathbb{R}^n$ is the parameter vector to be optimized and $z$ is a random vector that can be sampled to evaluate the loss $\ell(\theta, z)$. We assume that the considered problem is second-order differentiable. To simplify the notation, we use $\hat{\mathcal{L}}(\theta)$ to denote a sampled noisy evaluation of $\mathcal{L}(\theta)$. A step of SGD with learning rate $\mu$ and an optional positive definite preconditioner $P$ is given by:

$$\theta_{i+1} = \theta_i - \mu P \, \partial \hat{\mathcal{L}}(\theta) / \partial \theta \mid_{\theta=\theta_i} \tag{1}$$

where $i$ is the iteration index, $\mu > 0$ is the learning rate, and $P$ typically is a variable or adaptive preconditioner. Once the solution enters a basin of attraction centered at a local minimum $\theta^*$, we can approximate the iteration step in equation 1 as:

$$\theta_{i+1} - \theta^* \approx (I - \mu P \hat{H})(\theta_i - \theta^*) \tag{2}$$

where $\hat{H} = \frac{\partial^2 \hat{\mathcal{L}}(\theta)}{\partial \theta^T \partial \theta} \mid_{\theta=\theta^*}$ is the sampled Hessian at the local minimum. Conceivably, the eigenvalue spread of $P\hat{H}$ largely determines the speed of convergence of the quasi-linear system in equation 2. Nearly quadratic convergence is possible if we can find a good approximation for $H^{-1}$. However, $\hat{H}$ is a noisy Hessian and is not necessarily positive definite, even if the exact one at $\theta^*$, i.e., $H$, is.

### 2.2 THE PRECONDITIONER FITTING CRITERION

We adopt the preconditioner fitting criterion proposed in Li (2015). Let $\delta g$ be the perturbation of gradient associated with parameter perturbation $\delta\theta$. Then, the fitting criterion is:

$$c(P) = E_{\delta\theta}[\delta g^T P \delta g + \delta\theta^T P^{-1} \delta\theta] \tag{3}$$

With auto differentiation tools, we can replace the pair $(\delta\theta, \delta g)$ with $(v, \hat{H}v)$, where $v$ is a random vector, and $\hat{H}v$ is the noisy Hessian-vector product, which can be evaluated as cheaply as gradients. Criterion equation 3 only has one positive definite solution, $P = (H^2 + E_v[\epsilon^2])^{-\frac{1}{2}}$, even for indefinite $H$, where $\epsilon = \hat{H} - H$ is a stochastic noise term. This preconditioner automatically dampens gradient noise. It is worth noting that criterion equation 3 gives the same preconditioner used in equilibrated SGD (ESGD) (Dauphin et al., 2015) and AdaHessian (Yao et al., 2021) when $P$ is diagonal, i.e., $E[v \odot v] \oslash E[(\hat{H}v) \odot (\hat{H}v)]$, where $\odot$ and $\oslash$ denote element-wise product and division, respectively.

## 2.3 Preconditioners on Lie Groups

It is natural to fit the preconditioner on a Lie group for several reasons. First, by rewriting equation equation 1 as $P^{-\frac{1}{2}}\theta_{i+1} = P^{-\frac{1}{2}}\theta_i - \mu\partial\hat{\mathcal{L}}(\theta)/\partial(P^{-\frac{1}{2}}\theta)\mid_{\theta=\theta_i}$, it is clear that a preconditioned SGD is equivalent to SGD in a new set of coordinates defined by $\vartheta = P^{-\frac{1}{2}}\theta$. This coordinate change consists of rotations and scalings, i.e., operations on the orthogonal group $O(n)$ and the group of nonsingular diagonal matrices. We can represent this coordinate transform with matrix $Q^{-1}$ and, accordingly, $P = Q^T Q$. Thus, we pursue a variable $Q$ on the Lie group to fit it.

Second, PSGD can also be viewed as SGD with transformed features when the parameters to be learned are a list of affine transform matrices (Li, 2019). Specifically, the most commonly used feature transformations (e.g., whitening, normalization, and scaling) can be represented as matrices on the affine groups. For example, the popular batch normalization (Ioffe and Szegedy, 2015), layer normalization (Ba et al., 2016), and group normalization (Wu and He, 2018) can be represented as a sparse affine Lie group matrix where only the diagonal and last column can have nonzero values (Li, 2019) (See B). The decorrelated batch normalization (Huang et al., 2018) is related to the whitening affine preconditioner in Li (2019). Thus, the Lie group arises as a natural object to work with.

Lie groups have two properties that are particularly suitable for our task. Like any group, a specific Lie group preserves certain symmetries or invariances. For example, with $Q \in GL^+(n,\mathbb{R})$, the general linear group with positive determinant, $\vartheta$ and $\theta$ will always have the same orientation. This eliminates the need for damping, or similar remedies, to avoid degenerate solutions, since $P = Q^T Q$ is guaranteed to be invertible. The equivariance property of Lie groups further facilitates the preconditioner fitting. The same group generator, i.e., the one at the identity matrix, can be used to move a preconditioner on any point of the Lie group.

In fact, the preconditioner $P$ estimated by PSGD converges to the inverse of "absolute" Hessian regardless of the definiteness of Hessian. From this, one can show that the parameters converge following the established results in open literature. For more details and proof see A.

**Proposition 2.1.** *Assume that $H$ is invertible, and $dQ = -\mu\frac{\partial c}{\partial Q}$ or $\mathcal{E} = -\mu Q^T\frac{\partial c}{\partial Q}$. Then, $Q$ converges to $\pm|H|^{-0.5}$ with the learning rule equation 6 and a small enough positive step size $\mu$.*

**Corollary 2.1.1.** *Assume that $\mathcal{L}(\theta)$ is second order differentiable with absolute eigenvalues of the Hessian well bounded, i.e., $0 < l \le |\lambda(H)| \le u < \infty$. Then with PSGD, the loss drops at least with a linear rate, and the parameters converge at least linearly to the optimal solution $\theta^*$ if it exists.*

**Corollary 2.1.2.** *Assume that $\mathcal{L}(\theta)$ is $\alpha$-strongly convex and $\beta$-smooth function. Then with learning rate $\mu = \alpha/\beta$, PSGD recovers Newton's method with update rule of Eq. equation 1, and convergences quadratically to the optimal solution $\theta^*$ as $\mathcal{L}(\theta_{t+1}) - \mathcal{L}(\theta_t) \le -\frac{\alpha}{2\beta^2}\|\frac{\partial\mathcal{L}(\theta_t)}{\partial\theta_t}\|^2$.*

It is worth mentioning that no convergence rate beyond linear and quadratic are observed for non-convex and convex for first or second order stochastic optimization respectively. Proposition 2.1 and Corollary 2.1.1 & 2.1.2 (proved in A.1, A.2, & A.3) are not aimed to push the theoretical convergence limits. Instead, together, they investigate how PSGD recovers Newton's method.

The preconditioners proposed in Li (2019) can only be applied to a list of affine transform matrix parameters. Although many machine learning models exclusively consist of affine transforms and activation functions, this is not always the case. Additionally, it can be impractical to reparameterize many existing modules, such as a convolutional layer, into their equivalent affine transformation form. Hence, in this paper, we propose two types of novel general purpose preconditioners.

## 3 General Purpose Lie Group Preconditioners

### 3.1 Sparse Matrix-Free Preconditioners

Let us consider bijective mappings that take vectors in $\mathbb{R}^n$ and map them to other vectors in the same space, i.e., $T : \mathbb{R}^n \mapsto \mathbb{R}^n$. The following theorem gives one systematic way to construct such sparse matrix-free Lie group preconditioners.

**Claim 3.1.** *Let $K = \{\sigma_1, \ldots, \sigma_m\}$ be a subgroup of the permutation group $S_n$. Then, linear transform $T : \mathbb{R}^n \mapsto \mathbb{R}^n$, $T(x|a_1, \ldots, a_m) = \sum_{i=1}^{m} a_i \odot \sigma_i(x)$, forms a subgroup of $GL(n, \mathbb{R})$ parameterized with $\{a_1, \ldots, a_m\}$ if $T(\cdot|a_1, \ldots, a_m)$ is bijective, where both $a_i$ and $x$ are in $\mathbb{R}^n$.*

See proof of Claim 3.1 in Appendix B.

*Example 1*: the group of invertible diagonal matrices. We must have $K = \{e\}$ if $|K| = 1$, where $e$ is the identity element of $S_n$, i.e., $e(x) = x$. Then, $T$ simply has a diagonal matrix representation, i.e., $T(x|a_1) = \text{diag}(a_1)x$. Criterion equation 3 gives the preconditioner in ESGD (Dauphin et al., 2015) and AdaHessian (Yao et al., 2021) as a special case when $P$ is on this group.

*Example 2*: The group of "X-shape matrices." Let $K = \{e, \sigma_f\}$, where $\sigma_f$ denotes the flipping permutation. Then, we can show that

$$T(\cdot|a, b)T(\cdot|u, v) = T(\cdot|a \odot u + b \odot \sigma_f(v), a \odot v + b \odot \sigma_f(u))$$
$$T^{-1}(\cdot|a, b) = T(\cdot|\sigma_f(a) \oslash c, -b \oslash c)$$

where $c = a \odot \sigma_f(a) - b \odot \sigma_f(b)$. Clearly, such transforms form a Lie group if they are invertible, i.e., no element of $c$ is zero. The matrix representation of this $T$ only has nonzero diagonal and anti-diagonal elements, thus the name X-shape matrix (XMat). This becomes our minimal overhead general purpose preconditioner for PSGD.

*Example 3*: The butterfly matrix. For an even $n$, subgroup $K = \{e, s_{\frac{n}{2}}\}$ induces a Lie group whose representations are invertible butterfly matrices, where $s_{\frac{n}{2}}$ denotes circular shifting by $\frac{n}{2}$ positions. This group of matrices are the building blocks of the Kaleidoscope matrices (Dao et al., 2020).

Additionally, the group $GL(n, \mathbb{R})$ can be recovered by letting $K = \{e, s_1, \ldots, s_{n-1}\}$, where $s_i$ denotes circular shifting by $i$ positions. But, $GL(n, \mathbb{R})$ is too expensive for large scale problems. The group of diagonal matrices, i.e., the Jacobi preconditioner, is sparse but empirically shown to be less effective without the help of momentum for certain machine learning problems (Dauphin et al., 2015). We are mostly interested in the cases with $2 \leq |K| \leq 4$. These Lie groups are sparse enough, yet simple enough to derive their inverse manually, and at the same time can significantly accelerate the convergence of SGD by shortcutting gradients separated far away in positions.

## 3.2 LOW-RANK APPROXIMATION PRECONDITIONER

Low-rank approximation (LRA) is a standard technique for processing large-scale matrices. Commonly adopted forms of positive definite low-rank approximation, such as $P = \rho I + UU^T$, cannot always be factorized as $P = Q^T Q$ for certain Lie groups, where $\rho > 0$ is a small positive number. Additionally, this form of approximation is not effective for reducing eigenvalue spread. In many real-world problems, the Hessian has a few very large and very small eigenvalues, i.e., tails on both ends of the spectra (Sagun et al., 2016; 2017). However, all the eigenvalues of $P$ in this form are lower bounded by $\rho$, meaning that it can only fit one tail of the spectra when $\text{rank}(U) \ll n$.

For this reason, we propose a new low-rank approximation with form $Q = \rho(I + UV^T)$, where $\rho$ is not necessarily small nor positive, and $U$ and $V$ have $r$ columns with $r \ll n$. To justify this form of approximation, we need to establish two facts. First, it forms a Lie group. Second, $P = Q^T Q$ with this form can fit both tails of the spectra of Hessian, providing an accurate characterization of the curvature of a function, improving optimization algorithms, and assessing their robustness.

**Claim 3.2.** *Preconditioner $P = Q^T Q$ with $Q = \rho(I + UV^T)$ can have positive eigenvalues arbitrarily larger than $\rho^2$ and arbitrarily smaller than $\rho^2$ with proper $U$ and $V$.*

**Claim 3.3.** *If $\rho \neq 0$ and $(I + V^T U)^{-1}$ or $(I + U^T V)^{-1}$ exists, $A_V(\rho, U) = \rho(I + UV^T)$ defines a subgroup of $GL(n, \mathbb{R})$ parameterized with $\rho$ and $U$. Similarly, $A_U(\rho, V) = \rho(I + UV^T)$ defines another subgroup of $GL(n, \mathbb{R})$ parameterized with $\rho$ and $V$.*

See proofs of Claim 3.2 &3.3 in Appendix C.1 & C.1.

The form of $Q$ in Claim 3.2 is rather constrained as $\rho$ is a scalar. In practice, we replace $\rho$ with another Lie group matrix and define $Q$ as $Q = B(I + UV^T)$. In our implementations, we choose $B$ to be the group of diagonal matrix with positive diagonals and update $B$ along with $U$ and $V$ on two separate Lie groups. Note that now $B(I + UV^T)$ generally no longer forms a single Lie group.

## 4 PRACTICAL CONSIDERATIONS

Above-proposed preconditioners can be fit online by minimizing criterion equation 3 using gradient descent on Lie groups. Unlike traditional gradient descent, moving an object on a Lie group is achieved by multiplying it with $I + \mu G$, where $G$ is the group generator and $\mu$ is small enough such that $\|\mu G\| < 1$. This series of small movements trace a curve on the Lie group manifold, and $G$ is always in the tangent space of the group as the Lie algebra is closed. See D for details.

Note that optimizer damping is neither necessary nor generally feasible on a Lie group, although it is widely used in other second-order optimizers to avoid degenerate solutions. On one hand, by fitting $Q$ on a connected Lie group, $P = Q^T Q$ cannot be singular. On the other hand, damping could be incompatible with certain forms of Lie groups, as we may not always be able to find another $Q'$ on the same group such that $Q'^T Q' = Q^T Q + \lambda I$, where $\lambda > 0$. This eliminates the need for setting up a proper damping schedule. However, gradient clipping can be helpful in promoting stability. The quadratic approximation leading to the quasi-linear system equation 2 is only valid within a certain region around $\theta$. Thus, $\|\delta\theta\| = \mu\|P\partial\hat{\mathcal{L}}(\theta)/\partial\theta\|$ should be small enough such that $\theta + \delta\theta$ still locates in this trust region. We can adjust $\mu$ or clip $\|P\partial\hat{\mathcal{L}}(\theta)/\partial\theta\|$ to ensure that $\|\delta\theta\|$ is small enough.

Theoretically, one Hessian-vector product evaluation doubles the complexity of one gradient evaluation. In practice, we could update the curvature estimation with probability 0.1. The cost of updating $P$ is negligible for $r \leq 100$ compared with SGD. Then the per iteration complexity of PSGD becomes $1 + 2 \times 0.1 = 1.2$ times of that of SGD, as shown empirically in 11 and 7.

Lastly, the learning rate for parameter updating and step size for preconditioner fitting are naturally normalized to be in the range $(0, 1)$. We have found a step size of $0.01$ to be an effective initial rate, and our method is robust to a wide range of learning rates and weight decays. (see Appendix F.4)

For the LRA preconditioner, the gradients on the Lie groups are given by $0.5\nabla_B = \text{diag}[(Ph)h^T - v(P^{-1}v)^T]$, $0.5\nabla_U = [(Qh)(Qh)^T - (Q^{-T}v)(Q^{-T}v)^T]V$ and $0.5\nabla_V = [(Qh)(Qh)^T - (Q^{-T}v)(Q^{-T}v)^T]U$, respectively. Interested readers can refer to the appendices for details. To put these together, we summarize the proposed PSGD methods into Algorithms $1 \sim 3$ as below. For more on Algs see E.

---

**Algorithm 1:** PSGD Optimizer

Initialize $\theta_0$, $t \leftarrow 0$, $Q_0 \propto I$

**While** $\theta_t$ not converged

  $t \leftarrow t + 1$

  $g_t \leftarrow \nabla_\theta f_t(\theta_{t-1})$

  **If** $u < p$ with $u \sim \mathcal{U}(0, 1)$

    $h_t \leftarrow \nabla_\theta(v_t^T g_t)$, s.t. $v_t \sim \mathcal{N}(0, I)$

    **Update** $Q_t$ **via** $(v_t, h_t)$

    Q Update Step

  **Else**

    $Q_t \leftarrow Q_{t-1}$

  $\mathfrak{g}_t \leftarrow Q_t^T Q_t g_t$

  $\theta_t \leftarrow \theta_{t-1} - \mu_1 \mathfrak{g}_t$

---

**Algorithm 2:** UVd Q Update Step

$Ph = Q^T(Qh)$

$P^{-1}v = Q^{-1}(Q^{-T}v)$   via Woodbury identity 2x

$\nabla_d = (Ph) \odot h - v \odot (P^{-1}v)$

$d \leftarrow d - \mu_2 d \odot \nabla_d / \max(|\nabla_d|)$

**If** $u < 0.5$ with $u \sim \mathcal{U}(0, 1)$

  $\nabla_U = (Qh)(Qh)^T V - (Q^{-T}v)(Q^{-T}v)^T V$

  $U \leftarrow U - \mu_2 \|\nabla_U V^T\|^{-1} \nabla_U (I + V^T U)$

**Else**

  $\nabla_V = (Qh)(Qh)^T U - (Q^{-T}v)(Q^{-T}v)^T U$

  $V \leftarrow V - \mu_2 \|U\nabla_V^T\|^{-1}(I + VU^T)\nabla_V$

**Return** $Q = (I + UV^T)\text{diag}(d)$

---

**Algorithm 3:** XMat Q Update Step

$Q^{-T}v = (a \odot v - b \odot v) \oslash (a \odot a - b \odot b)$

$\nabla_a = (Qh) \odot (Qh) - (Q^{-T}v) \odot (Q^{-T}v)$

$\nabla_b = (Qh) \odot (\overline{Qh}) - (Q^{-T}v) \odot (\overline{Q^{-T}v})$

**If** $b$ has odd length

  Set central element of $\nabla_b$ to zero

$\mu = \frac{\mu_2}{\max[\max(|\nabla_a|), \max(|\nabla_b|)]}$

$a \leftarrow a - \mu(\nabla_a \odot a + \nabla_b \odot \bar{b}))$

$b \leftarrow b - \mu(\nabla_a \odot b + \nabla_b \odot a)$

**Return** $Q_t \leftarrow \text{diag}(a) + \text{adiag}(b)$

---

## 5 EMPIRICAL RESULTS

In this work, we evaluate the performance of the PSGD algorithm on a diverse set of tasks. First we consider two toy problems, Rosenbrock objective minimization to show quadratic convergence of PSGD, as well as using a LeNet5 (Lecun et al., 1998) (see Figure 1b & Table 9) for MNIST (LeCun and Cortes, 2010) digit recognition for studying the generalization property of PSGD.

Next, we benchmark more large-scale vision, natural language processing (NLP), and reinforcement learning (RL) tasks. For each task we benchmark PSGD vs the leading SoTA optimizers. In the domain of computer vision, we evaluate algorithm performance on the MNIST dataset via convex large-scale logistic regression (see Appendix F.5). Additionally we consider the CIFAR10 (CF10) (Krizhevsky, 2009) and CF10 derived datasets, namely noisy labels, blur-deficit, adversarial attacks, and class imbalances, (see Figure 2a & Table 11) with ResNet18 (RN18) to explore generalization performance. For NLP tasks, we study the use of LSTMs (Hochreiter and Schmid-

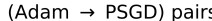

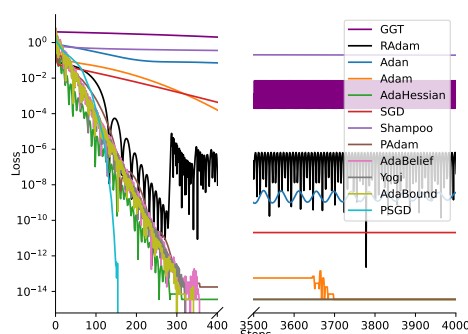
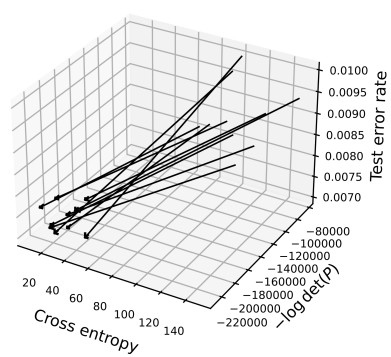

(a) Rosenbrock objective minimization     (b) LeNet5 minima pair Adam→PSGD

Figure 1: (a) Rosenbrock objective minimization comparison among PSGD and its competitors. Only PSGD shows a clear quadratic convergence curve. (b) MNIST hand written digit recognition with LeNet5. Hessians at the minima of Adam are estimated with a dummy LRA PSGD optimizer that only updates the preconditioner.

huber, 1997) and GPT-2 style transformers (Radford et al., 2019), on various text corpora, including the Penn Treebank (Marcus et al., 1993), the complete works of Shakespeare, and the Open Web Text (OWT) dataset Gokaslan and Cohen (2019) (see Table 14 & 1). In the RL setting, we consider a Proximal Policy Optimization (PPO) (Schulman et al., 2017) applied to the HalfCheetah and RoboWalker environments using OpenAI's gym environments (Brockman et al., 2016) (see Fig. 3).

To provide insight into the fundamental differences between SGD and PSGD, we perform uncertainty and forgettability analysis (Toneva et al., 2018). Finally, we examine the pathological delayed XOR problem, first introduced in Hochreiter and Schmidhuber (1997), in order to further our understanding of the strengths and limitations of different optimization methods.

## 5.1 Performance Study with Toy Examples

The first toy example is the minimization of the Rosenbrock function demonstrating the recovery of curvature information by PSGD. As shown by Proposition 2.1, preconditioner $P$ follows $(H^T H)^{-1/2}$. This property helps PSGD to escape saddle points as $P \to \infty$ when $H \to 0$, and accelerate convergence around the basin of attraction. This quadratic convergence behavior of PSGD is clearly shown by the convergence curve solving the Rosenbrock benchmark problem in Fig. 1a.

The second toy example demonstrates that PSGD preserves the generalization property of its kin, SGD, as the gradient noises regularize both parameter and preconditioner updates. The task is the MNIST digit recognition with the LeNet5. Adam is selected as the counter-example as it is known to be more easily trapped in sharp minima than SGD (Zhou et al., 2020b). Fig. 1b shows ten pairs of minima, each starting from the same random initial initialization. We see that PSGD converges to minima with flatter or smaller Hessian, i.e., larger preconditioner. From the view of information theory, the total cross entropy and $0.5 \log \det(H) \approx -0.5 \log \det(P)$ are good proxies of the description lengths of the train image-label pairs and model parameters, respectively. Fig. 1b shows that minima with smaller description lengths tend to perform better on the test sets as well, as suggested by an arrow pointing to the down-left-front corner of the cube.

## 5.2 CIFAR-10 and Friends on ResNet18

We train an RN18 on the CF10 image classification task using PSGD, SGD, Adam, and Apollo (Adaptive Quasi Newton Diagonal). We adopt the cosine learning rate scheduler (Loshchilov and Hutter, 2016) and train for 200 epochs (see F.7). We find that as other first and second-order optimizers performance reduces and increases in variance as task complexity increases, PSGD is able to sustain the classification accuracy of the RN18, outperforming other optimizers by as much as much as 64%, see Figure 2a Table 11 for details.

These tasks provide a rigorous test for the generalization ability of optimizers. We maintain the tuned hyperparameters from the standard RN18 CF10 classification task for a fair comparison between the optimizers. For robustness & sensitivity analysis on PSGD's hyper-parameters as well as more experimental details see F.4. We update the preconditioner of PSGD every ten iterations, resulting in a 20% overhead over SGD, see Table 11 and F.3 for empirical and theoretical timing analysis.

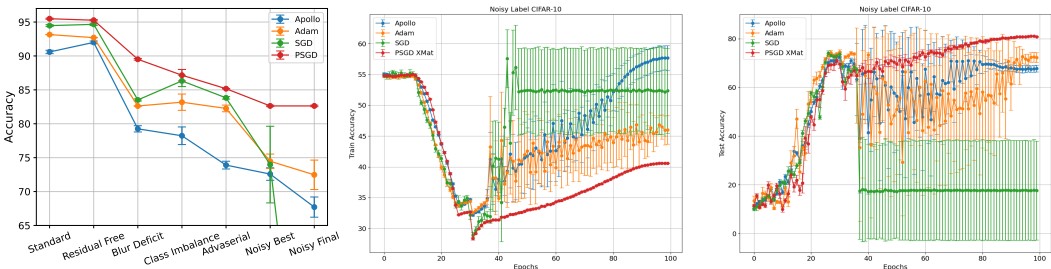

(a) CF10 RN18 Experiments     (b) Train Acc: Noisy Label CF10     (c) Test Acc: Noisy Label CF10

Figure 2: CIFAR-10 ResNet-18: (a) Robustness of PSGD: We clearly see as classification task increases in complexity($\rightarrow$), PSGD is able to consistently outperform other first and second order optimizers. (b) Assym Label Noise Train Acc: Accuracy plots based on incorrect noisy labels. PSGD effectively mitigates label noise, learning the true underlying solution with low variance, while other optimizers tend to overfit/memorize the miss-leading trainset. (c) Assym Label Noise Test Acc: Under ground truth test labels, we see that PSGD reaches a significantly better test accuracy with a very low variance compared to Apollo, Adam, and SGD.

**CIFAR10 with Asymmetric Noisy Labels**

We asymmetrically perturb $60\%$ of the CF10 training labels randomly, resulting in one of the classes having approximately $55\%$ incorrect labels and the other 9 classes having $5\%$ incorrect labels, yielding *asymmetric label noise*. We use (P)SGD, Adam & Apollo to train an RN18 on this task for 100 epochs with 8 different seeds and compare train and test classification accuracy Fig. 4a & 4b.

Looking at Figure 4a & 4b, we see that PSGD achieves the lowest average train accuracy (assuming incorrect noisy labels) with the highest ground truth test accuracy. SGD gets an average training accuracy between Adam and Apollo. SGD has seven runs reaching $55\%$ memorizing the train set, yielding $10\%$ accuracy on the test set. And one run (due to *lucky initialization*) reaching $34\%$ on the train set, learning an underfit yet generalizing solution, yielding $77\%$ on the test set.

For SGD, this is not a standard case of over-fitting nor a case of catastrophic forgetting, since the training accuracy does not change. Instead, our experiments show there exists a tipping point in the test accuracy at around epoch 35, where within a single epoch the test accuracy drops from $71\%$ to $10\%$ while the training accuracy shows no intelligible indication of this change. Furthermore, the test accuracy does not increase for the rest of the 65 epochs. At the 35 epoch mark Adam and Apollo both go through a period of high variance over-fitting that eventually converges. Note that the average margins or predicted probabilities in Table 12 indicate that PSGD finds a generalizable solution whereas other first and second-order optimizers fall into a regime of overfitting/memorization further discussed in 5.5. In conclusion, we show PSGD finds a generalizable solution that can mitigate both over and underfitting. Other optimizers easily over/under-fit or memorize incorrect image label pairs, have large optimization instabilities during training, and reach degenerate solutions where the NNs have reasonable train accuracy but random test accuracy.

For more details and experiments on learning under *asymmetric* and *symmetric* label noise see F.1 and F.1.

## 5.3 LANGUAGE MODELING

Transformers have become the de facto language modeling architecture, largely replacing RNN-based models. Transformers employ sparse self-attention mechanisms that allow for efficient computation of gradients. Thus, they are typically trained using first-order methods. In this section, we study the effect of curvature information in training transformer models. Additionally, we provide results for LSTM-based experiments for predicting the Penn TreeBank Dataset using Zhuang et al. (2020)'s framework in Table 14.

**nanoGPT** In a recent study, Karpathy (2022) proposed a framework for reproducing GPT-2 results using the OpenWebText dataset. Here, we expand upon this by investigating the training of GPT-2 models of different sizes on both the OpenWebText and Shakespeare character datasets. Our primary objective is to benchmark the performance of PSGD against other SoTA optimization algorithms.

As shown in Table 1, our results indicate that PSGD consistently outperforms other optimization algorithms across various model sizes. Notably, a moderate gap in perplexity is observed for the

Table 1: Comparing different test loss of optimizers over GPT-2 style transformers on the Shakespeare-char (SC) and OpenWebText (OWT) datasets. PSGD outperforms other optimizers including the SoTA optimizer AdamW over various model sizes. Trainined on a single NVIDIA 3080 GPU.

| nanoGPT | PSGD | SGD | AdamW | AdanW | AdaBelief | AdanBelief |
|---|---|---|---|---|---|---|
| SC: 0.82M | **4.52** | 4.68 | 4.68 | 5.52 | 5.94 | 6.66 |
| SC: 1.61M | **4.47** | 4.75 | 5.03 | 5.054 | 5.06 | 6.47 |
| SC: 6.37M | **4.53** | 5.31 | 19.53 | 4.92 | 21.04 | 5.34 |
| OWT: 50M | **250.7** | | 591.8 | | | |

smaller GPT-2 model trained for 5k iterations on the Shakespeare-char dataset, with a significantly larger gap observed on the 50M parameter GPT-2 LLM trained on the OpenWebText dataset, for 600k iterations using AdamW and 100k iterations using PSGD. We found that decreasing the *precond lr* to 0.001 greatly improved the performance of transformer models. Lowering the *precond lr* smoothens the curvature in the sparse embedding layer of GPT-2 over time and enables the optimizer to consider a larger window of curvature information. This "curvature memory" improved performance and prevented divergence resulting from the otherwise sparse embedding space.

## 5.4 REINFORCEMENT LEARNING

Here we consider two standard Proximal Policy Optimization (PPO) problems in Reinforcement Learning (RL): Walker2d and HalfCheetah. We compare the performance of the two SOTA optimizers AdaBelief and Adan to PSGD. We optimize the actor and critic independently. We find that PSGD can find a higher reward for both Walker2d & HalfCheetah as shown in Figure 3.

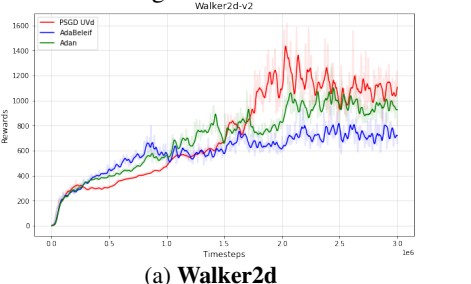 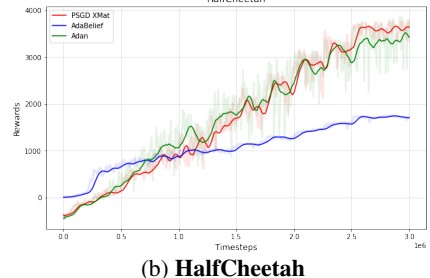

(a) **Walker2d**    (b) **HalfCheetah**

Figure 3: PSGD outperforms SOTA optimizers on PPO RL on Walker2d & HalfCheetah.

## 5.5 TOWARDS UNDERSTANDING SECOND ORDER OPTIMIZERS

With the performance distinctions between PSGD and the SoTA optimiziers now apparent, we aim to conduct a series of simple experiments to better understand the nature of solutions PSGD finds.

**Uncertainty Analysis** We train a standard RN18 on CF10 with PSGD, SGD, Adam, and Apollo, and after 200 epochs we check the entropy over the softmax-logits and miss-classification margin Toneva et al. (2018) of both NNs. We see in Table 12 that PSGD has higher entropy and a lower margin of classification compared to other optimizers. This very low minimum entropy of other optimizers may be interpreted as a form of overfitting or memorization. Intuitively, some data points that are low entropy have information that is easily memorized by NNs, giving up network capacity learning points that do not improve generalization. Conversely, we observe that PSGD never becomes overly certain about any data point, with the minimum entropy being six orders of magnitude larger than other optimizers. Similar trends can be observed in the mean and maximum entropy values. From these statistics, we believe standard first and second-order optimizers can push NNs into a dangerous regime of overconfidence which given clean labels can reduce their generalization capacity with the potential of catastrophic forgetting. In the scenario where a network is given noisy labels (or imaginably poisoned points), training other SoTA methods may lead to memorization of the training set with little to no generalization capacity as seen in the noisy labeled experiments 2a.

PSGD's higher entropy and lower margins are indications of a larger exploration of the parameter space, more diverse solutions, and a lack of memorization. Suggesting that PSGD is able to better balance the trade-off between overfitting and underfitting, without memorizing points.

For more on the nature of low and high entropy points in a dataset see F.6 & F.6.

**Forgettability Statistics and Learning** We revisit Toneva et al. (2018)'s forgettability experiments, which found that one can prune 30% of the CIFAR-10 train samples without loss of generalization. The study found that a point's utility for generalization increases as it is learned and then

Table 2: Uncertainty Statistics: PSGD's higher uncertainty leads to better generalization and less over-fitting.

| NTK | Entropy | | | Margin | | |
|---|---|---|---|---|---|---|
| | Min | Mean | Max | Min | Mean | Max |
| PSGD | 0.139 | 0.260 | 1.545 | 0.144 | 0.956 | 0.994 |
| SGD | $7 \times 10^{-7}$ | 0.01 | 0.8975 | 0.3925 | 0.999 | 1 |
| Adam | $1.5 \times 10^{-7}$ | 0.009 | 0.8645 | 0.3625 | 0.999 | 1 |
| Apollo | $1 \times 10^{-6}$ | 0.05 | 0.8851 | 0.4326 | 0.999 | 1 |

Table 3: Forgetting statistics for CIFAR10 on ResNet18. PSGD finds better forgettability statistics outperforming SGD.

| Forgetting | 50k | 25k | 15k | 5k |
|---|---|---|---|---|
| PSGD | 96.65 | 95.83 | 94.95 | 56.46 |
| SGD | 96.21 | 95.56 | 93.7 | 42.48 |

subsequently forgotten during training, regardless of the optimizer or architecture used. Essentially, forgettability ordering shows which data points an NN uses to define high-dimensional boundaries akin to support vectors. We investigate whether the performance difference between PSGD and SGD's generalization performance can be attributed to this forgettability ordering.

We train the RN18 four times, keeping the top $N$ important points based on each optimizer's expected forgettability score. Table 13 shows that PSGD focuses on points that are central to generalization. We see this since, when we limit the dataset to only the 5k most forgettable data points deemed by each optimizer, we see PSGD is able to outperform SGD by nearly 14%.

For more insight on forgettability statistics and its connection to entropy see F.6, F.6, and F.7.

**PSGD Preserves Neuro-Plasticity** Recently, Achille et al. (2017) studied the phenomenon of neuroplasticity in NNs. They found that NNs exhibit a critical learning period, during which if a learning deficit is not removed, the NN becomes unable to learn effectively. To simulate this deficit, CIFAR10 images are blurred for the first half of training, after which the deficit is removed. Following Achille et al. (2017), we used an exponential learning rate decay. To investigate the impact of optimization on neuro-plasticity we compare SGD and PSGD. We find that PSGD retains neuro-plasticity outperforming SGD by 6% and 3% on test and train sets seen in Fig 2a, 14b and Table 11.

For more on critical learning periods and the importance of curvature information early in training for improving distributional shift for transfer learning, see and F.7 and F.7.

**Understanding long term dependence and memorization via the Delayed XOR Problem** The task is to predict the XOR relation of $a$ and $b$ randomly scattered far away in a long sequence. This problem is challenging for many architectures and optimizers because it cannot be "partially solved" as memorizing either $a$ or $b$ alone does not help to predict $XOR(a, b)$. We consider solving it with the vanilla RNNs and LSTMs optimized using SGD, AdaBelief, Adan, AdaHessian, Apollo, Hessian Free, Shampoo, KFAC, and PSGD at different sequence lengths. Apollo was not able to solve the XOR problem in any scenario. The success rates for each optimizer are shown in Table 4. Clearly, PSGD LRA is the only method that can solve the XOR problem passing sequence length 32 with an RNN. Furthermore, LSTMs show no benefit to RNNs without using curvature information. Also, RNN optimized with PSGD outperforms LSTM optimized by any other methods. These results hint at two points. First, the choice of optimizer could be equally important as model architecture. Second, similar to the CIFAR10 tasks, PSGD relies less on the memorization of train samples in problem-solving. See F.7 for rank analysis.

Table 4: Success rate on the delayed XOR problem with variant sequence lengths, optimizers and networks.

| XOR | PSGD LRA | | AdaHessian | | KFAC | | HessianFree | | Shampoo | | SGD | | AdaBelief | | Adan | |
|---|---|---|---|---|---|---|---|---|---|---|---|---|---|---|---|---|
| Length | RNN | LSTM | RNN | LSTM | RNN | LSTM | RNN | LSTM | RNN | LSTM | RNN | LSTM | RNN | LSTM | RNN | LSTM |
| 32 | 1 | 1 | 1 | 1 | 1 | 1 | 1 | 1 | 0 | 1 | 1 | 1 | 1 | 1 | 1 | 1 |
| 55 | 1 | 1 | 0 | 1 | 0 | 0 | 0 | 0.6 | 0 | 0.8 | 0 | 0 | 0 | 0 | 0 | 0 |
| 64 | 1 | 1 | 0 | 0.8 | 0 | 0 | 0 | 0 | 0 | 0.6 | 0 | 0 | 0 | 0 | 0 | 0 |
| 112 | 1 | 1 | 0 | 0 | 0 | 0 | 0 | 0 | 0 | 0 | 0 | 0 | 0 | 0 | 0 | 0 |

# 6 CONCLUSION

In conclusion, this work has presented a comprehensive study of the proposed general-purpose Lie group preconditioners for the optimization of deep learning problems. We have provided theoretical guarantees for the convergence of Lie group preconditioned optimization methods, and empirical results demonstrating PSGD outperforming SoTA optimizers in generalization, robustness, and stability across various tasks in Vision, NLP, and RL. Furthermore, our analysis of forgettability and entropy shows that PSGD effectively focuses on forgettable points, leading to improved generalization. These findings provide valuable insights for further developing efficient and effective optimization techniques for deep learning models.

## 7    REPRODUCIBILITY

We ran all our code with reproducibility in mind. We have kept seeds as well as hyper-parameters for each experiment and will release them to public once published. We have included a codebase as a zip for now to retain anonymity. Note all results have been averaged over 8-16 experiments with variances provided. Also for PSGD we have hyper-parameter sweeps to show that we are robust to changes in these values (see Table F.4).For all Propositions, Corollarys and Claims in the main we provide proofs with all assumptions states in the appendix (see Appendix A, B, and C). Additionally, since Lie Groups are not a ubiquitous framework for machine learning, we provide both crucial as well as extra math for those interested (see D.1 and G). Furthermore, we provide many practical considerations and limitations that come-about in implementation G.2.

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

# A ON THE CONVERGENCE OF PSGD

In this section, we first show that the preconditioner $P$ estimated by PSGD converges to the inverse of Hessian under mild conditions. We then show the quadratic convergence of PSGD under the convex setting.

## A.1 PSGD'S PRECONDITIONER $P$ RECOVERS $H^{-1}$

To begin, let us formulate the problem clearly. Let $(v, h)$ be a pair of vector and the associated noisy Hessian-vector product. We assume that $v$ is drawn from distribution $\mathcal{N}(0, I)$. The noisy Hessian-vector product $h$ is modeled by

$$h = H_0 v + \varepsilon$$

where $H_0$ is the true Hessian, and $\varepsilon$ is a noise term independent of $v$ due to use of sample averaged loss instead of the true loss. Note, the full Hessian expansion is circumvented, and instead only requires us to calculate the much cheaper Hessian vector product $H_0 v$. Then, we can write the criterion for the preconditioner estimation in PSGD as

$$
\begin{aligned}
c(P) =& E[h^T P h + v^T P^{-1} v] \\
=& \mathrm{tr}(P E[hh^T] + P^{-1} E[vv^T]) \\
=& \mathrm{tr}(PH^2 + P^{-1})
\end{aligned}
\tag{4}
$$

where

$$H^2 = H_0^2 + E[\varepsilon \varepsilon^T]$$

Clearly, $H^2$ is positive semi-definite regardless of the definiteness of $H_0$.

Note, we do not assume any definiteness or sparsity properties for $H_0$. Hence, in general, we assume that $Q$ is fitted on a connected branch of the general linear group with learning rule

$$Q^{\mathrm{new}} = Q^{\mathrm{old}} + dQ = Q^{\mathrm{old}} + Q\mathcal{E} \tag{5}$$

where $Q$ relates to $P$ as

$$P = Q^T Q,$$

and both $dQ$ and $\mathcal{E}$ are sufficiently small matrices related by $dQ = Q\mathcal{E}$. One technical difficulty in proving the convergence of $Q$ with the learning rule equation 5 is that it cannot exclude any rotation ambiguity of $Q$ as suggested by

$$(UQ)^T(UQ) = Q^T(U^T U)Q = Q^T Q$$

where $U$ can be any matrix on the orthogonal group. To ameliorate this technical issue, we constrain $dQ$ to take on certain structure. In our proof, we assume that both $Q^{\mathrm{old}}$ and $dQ$ are symmetric such that $Q^{\mathrm{new}}$ is always symmetric as well. To achieve such a constraint, we should update $Q^{\mathrm{old}}$ on the Lie group as

$$
\begin{aligned}
Q' =& Q^{\mathrm{old}} + Q\mathcal{E} \\
Q^{\mathrm{new}} =& [(Q')^T Q']^{0.5}
\end{aligned}
\tag{6}
$$

In this way, $dQ = Q^{\mathrm{new}} - Q^{\mathrm{old}}$ is guaranteed to be symmetric as long as the starting guess $Q^{\mathrm{old}}$ is symmetric as well. Still, in practice we have used the learning rule equation 5, and equation 6 only serves for the purpose of proof.

**Proposition 2.1.** *Assume that $H$ is invertible, and $dQ = -\mu \frac{\partial c}{\partial Q}$ or $\mathcal{E} = -\mu Q^T \frac{\partial c}{\partial Q}$. Then, $Q$ converges to $\pm |H|^{-0.5}$ with the learning rule equation 6 and a small enough positive step size $\mu$.*

*Proof.* Given a small perturbation $dQ$ of $Q$, the change of $P$ is given by

$$
\begin{aligned}
dP =& (Q + dQ)^T (Q + dQ) - Q^T Q \\
=& Q^T dQ + dQ^T Q + dQ^T dQ
\end{aligned}
$$

The change of $P^{-1}$ is a little more complicated. We omit any terms smaller than $\mathcal{O}[(dQ)^2]$, and can expand $dP^{-1}$ as below

$$
\begin{aligned}
dP^{-1} =& (P + dP)^{-1} - P^{-1} \\
=& -P^{-1}dPP^{-1} + P^{-1}dPP^{-1}dPP^{-1} \\
=& \left( -Q^{-1}dQP^{-1} - P^{-1}dQ^TQ^{-T} - P^{-1}dQ^TdQP^{-1} \right) \\
& + \left( Q^{-1}dQQ^{-1}dQP^{-1} + Q^{-1}dQP^{-1}dQ^TQ^{-T} \right. \\
& \left. + P^{-1}dQ^TdQP^{-1} + P^{-1}dQ^TQ^{-T}dQ^TQ^{-T} \right) \\
=& -Q^{-1}dQP^{-1} - P^{-1}dQ^TQ^{-T} + Q^{-1}dQQ^{-1}dQP^{-1} \\
& + Q^{-1}dQP^{-1}dQ^TQ^{-T} + P^{-1}dQ^TQ^{-T}dQ^TQ^{-T}
\end{aligned}
$$

Now, the change of the PSGD criterion is given by

$$
\begin{aligned}
dc =& \mathrm{tr}(dPH^2 + dP^{-1}) \\
=& \mathrm{tr}(H^2Q^TdQ + H^2dQ^TQ + H^2dQ^TdQ) \\
& + \mathrm{tr}(-Q^{-1}dQP^{-1} - P^{-1}dQ^TQ^{-T} + Q^{-1}dQQ^{-1}dQP^{-1} \\
& + Q^{-1}dQP^{-1}dQ^TQ^{-T} + P^{-1}dQ^TQ^{-T}dQ^TQ^{-T}) \\
=& 2\mathrm{tr}(H^2Q^TdQ - P^{-1}Q^{-1}dQ) \\
& + \mathrm{tr}(dQ^TdQH^2 + 2dQQ^{-1}dQP^{-1}Q^{-1} + dQP^{-1}dQ^TQ^{-T}Q^{-1})
\end{aligned}
\tag{7}
$$

From equation 7, the first order derivatives of $c$ with respect to $Q$ is given by

$$
\frac{\partial c}{\partial Q} = QH^2 - Q^{-T}P^{-1}
$$

A stationary point is obtained by letting $\frac{\partial c}{\partial Q} = 0$, which leads to $H^2 = P^{-2}$. Hence, such a $P$ always exists as long as $H^2$ is invertible. Via relationship $dQ = Q\mathcal{E}$, the gradient on the Lie group is

$$
\nabla_{\mathcal{E}} = Q^T \frac{\partial c}{\partial Q}
$$

Thus, fitting $Q$ on the Lie group yields the same stationary point since $Q$ is invertible. Note that the stationary points only tell us that $Q^TQ = H^{-1}$ without excluding the rotation ambiguity.

Without the constraint $dQ = dQ^T$, the second order derivative of $c$ with respect to $Q$ can be shown to be

$$
\frac{\partial^2 c}{\partial(\mathrm{vec}(Q))^2} = 2I \otimes H^2 + 2J^T[Q^{-1} \otimes (QP)^{-T}] + 2[Q^{-T} \otimes (QP)^{-1}]J + 2(QQ^T)^{-1} \otimes P^{-1}
$$

where $J$ is the permutation matrix satisfying

$$
\mathrm{vec}(dQ^T) = J\,\mathrm{vec}(dQ)
$$

Via relationship $dQ = Q\mathcal{E}$, the second order derivative on the Lie group is

$$
\nabla_{\mathcal{E}}^2 = (Q^T \otimes I)\frac{\partial^2 c}{\partial(\mathrm{vec}(Q))^2}(Q \otimes I)
\tag{8}
$$

We see that neither $\frac{\partial^2 c}{\partial(\mathrm{vec}(Q))^2}$ nor $\nabla_{\mathcal{E}}^2$ is guaranteed to be positive definite. Hence, the learning rule equation 5 does not convergence to a point as expected due to the rotation ambiguity.

On the other hand, both $Q$ and $dQ$ are symmetric with the learning rule equation 6. Thus, the second order derivative of $c$ with respect to $Q$ simplifies to

$$
\frac{\partial^2 c}{\partial(\mathrm{vec}(Q))^2} = 2I \otimes H^2 + 2Q^{-1} \otimes (QP)^{-T} + 2Q^{-T} \otimes (QP)^{-1} + 2(QQ^T)^{-1} \otimes P^{-1}
$$

At a stationary point, we have $Q = \pm P^{0.5} = \pm|H|^{-0.5}$. Thus, this second order derivative reduces to

$$\frac{\partial^2 c}{\partial(\text{vec}(Q))^2}\big|_{Q=\pm|H|^{-0.5}} = 2I \otimes H^2 + 4H^{0.5} \otimes H^{1.5} + 2H \otimes H \tag{9}$$

By equation 8, the second order derivative on the Lie group is

$$\nabla^2_{\mathcal{E}}\big|_{Q=\pm|H|^{-0.5}} = 2H^{-1} \otimes H^2 + 4H^{-0.5} \otimes H^{1.5} + 2I \otimes H \tag{10}$$

Now, both $\frac{\partial^2 c}{\partial(\text{vec}(Q))^2}$ and $\nabla^2_{\mathcal{E}}$ are positive definite for an invertible $H$ at the stationary point. Thus, $Q$ converges to either stationary point, i.e., $|H|^{-0.5}$ or $-|H|^{-0.5}$. $\qquad\square$

From Proposition 2.1, we see that $P$ converges to $|H_0|^{-1}$ when $\varepsilon = 0$, i.e., inverse of the "absolute" Hessian. With stochastic gradient noises, $\varepsilon \neq 0$ and we always have $P \prec |H_0|^{-1}$. This is not a bug but rather a feature of PSGD that damps the gradient noises perfect such that $PE[\delta g \delta g^T]P = E[\delta\theta\delta\theta^T]$ Li (2015), a relationship generalizing the Newton method to non-convex and stochastic optimizations. Unlike the damping strategies in other second order methods, this built-in gradient noise damping mechanics of PSGD does not requires any tuning effort.

## A.2 Linear Convergence of PSGD under General Setting

**Corollary 2.1.1.** *Assume that $\mathcal{L}(\theta)$ is second order differentiable with absolute eigenvalues of the Hessian well bounded, i.e., $0 < l \leq |\lambda(H)| \leq u < \infty$. Then with PSGD, the loss drops at least with a linear rate, and the parameters converge at least linearly to the optimal solution $\theta^*$ if it exists.*

*Proof.* For a general nonconvex problems, we assume that the eigenvalues of Hessian is well bounded as

$$0 < l \leq |\lambda(H)| \leq u < \infty$$

Then by Proposition 2.1, $P$ converges to $|H|^{-1}$. Thus

$$0 < 1/u \leq \lambda(P) \leq 1/l < \infty$$

The learning rule for parameters with positive step $\mu$ and preconditioner $P$ follows as

$$\begin{aligned}
d\mathcal{L}(\theta) &= (d\theta)^T \frac{\partial\mathcal{L}(\theta)}{\partial\theta} \\
&= -\mu P \left(\frac{\partial\mathcal{L}(\theta)}{\partial\theta}\right)^T \frac{\partial\mathcal{L}(\theta)}{\partial\theta} \\
&= -\mu\text{tr}\left[\frac{\partial\mathcal{L}(\theta)}{\partial\theta} P \left(\frac{\partial\mathcal{L}(\theta)}{\partial\theta}\right)^T\right] \\
&\leq -\frac{\mu}{u}\text{tr}\left[\frac{\partial\mathcal{L}(\theta)}{\partial\theta} \left(\frac{\partial\mathcal{L}(\theta)}{\partial\theta}\right)^T\right] \\
&= -\frac{\mu}{u}\left\|\frac{\partial\mathcal{L}(\theta)}{\partial\theta}\right\|^2
\end{aligned}$$

Thus the loss decreases at least with a linear rate. Convergence to a local minimum, if exists, with at least a linear rate immediately follows from the convergence of loss as the Hessian is assumed to be nonsingular everywhere. $\qquad\square$

## A.3 Quadratic Convergence of PSGD under Convex Setting

In the previous subsection we proved that the estimate of the preconditioner $P$ recovers the true inverse Hessian. As such, under the assumptions detailed in 2.1.2 bellow, PSGD recovers Newton's method.

**Corollary 2.1.2.** *Assume that $\mathcal{L}(\theta)$ is $\alpha$-strongly convex and $\beta$-smooth function. Then with learning rate $\mu = \alpha/\beta$, PSGD recovers Newton's method with update rule of Eq. equation 1, and convergences quadratically to the optimal solution $\theta^*$ as $\mathcal{L}(\theta_{t+1}) - \mathcal{L}(\theta_t) \leq -\frac{\alpha}{2\beta^2}\|\frac{\partial\mathcal{L}(\theta_t)}{\partial\theta_t}\|^2$.*

*Proof.* We prove Corollary 2.1.2 (similarly to the proof of Newton's method in Boyd and Vandenberghe (2004)) for the following general update rule: Note from Proposition 2.1, we have that $P$ converges to $|H_0|^{-1}$ when $\varepsilon = 0$, i.e., inverse of the "absolute" Hessian. With stochastic gradient noises, $\varepsilon \neq 0$ and we always have $P \prec |H_0|^{-1}$.

$$\Delta\theta_t = \mathbf{H}_t^{-1}\mathbf{g}_t \tag{11}$$

$$\theta_{t+1} = \theta_t - \mu\Delta\theta_t \tag{12}$$

Define $\lambda(\theta_t) = (\mathbf{g}_t^T\mathbf{H}_t^{-1}\mathbf{g}_t)^{1/2}$. Since $\mathcal{L}(w)$ is $\beta$-smooth, we have

$$\mathcal{L}(\theta_{t+1}) \leq \mathcal{L}(\theta_t) - \mu\mathbf{g}_t^T\Delta\theta_t + \frac{\mu^2\beta\|\Delta\theta_t\|^2}{2} \tag{13}$$

$$\leq \mathcal{L}(\theta_t) - \mu\lambda(\theta_t)^2 + \frac{\beta}{2\alpha}\mu^2\lambda(\theta_t)^2, \tag{14}$$

where in the last equality, we used

$$\lambda(\theta_t) = \Delta\theta_t\mathbf{H}_t\Delta\theta_t^T. \tag{15}$$

Therefore, using step size $\hat{\mu} = \frac{\alpha}{\beta}$ we have $\theta_{t+1} = \theta_t - \hat{\mu}\Delta\theta_t$

$$\mathcal{L}(\theta_{t+1}) \leq \mathcal{L}(\theta_t) - \frac{1}{2}\hat{\mu}\lambda(\theta_t)^2 \tag{16}$$

Since $\alpha I \preceq \mathbf{H}_t \preceq \beta I$, we have

$$\lambda(\theta_t)^2 = \mathbf{g}_t^T\mathbf{H}_t^{-1}\mathbf{g}_t \geq \frac{1}{\beta}\|\mathbf{g}_t\|^2, \tag{17}$$

and therefore $\mathcal{L}$ decreases as follows,

$$\mathcal{L}(\theta_{t+1}) - \mathcal{L}(\theta_t) \leq -\frac{1}{2\beta}\hat{\mu}\|\mathbf{g}_t\|^2 = -\frac{\alpha}{2\beta^2}\|\mathbf{g}_t\|^2. \tag{18}$$

$\square$

## B  CONSTRUCTION OF MATRIX-FREE PRECONDITIONER

The following statement gives a systematic way to construct a family of black-box matrix-free preconditioners.

**Claim 3.1.** *Let $K = \{\sigma_1, \ldots, \sigma_m\}$ be a subgroup of the permutation group $S_n$. Then, linear transform $T : \mathbb{R}^n \mapsto \mathbb{R}^n$, $T(x|a_1, \ldots, a_m) = \sum_{i=1}^{m} a_i \odot \sigma_i(x)$, forms a subgroup of $GL(n, \mathbb{R})$ parameterized with $\{a_1, \ldots, a_m\}$ if $T(\cdot|a_1, \ldots, a_m)$ is bijective, where both $a_i$ and $x$ are in $\mathbb{R}^n$.*

*Proof.* The proof follows by showing that such matrices have the following four properties required to form a Lie group.

First, we show that $I$ is the identity element. Note that $K$ has element $e$ since it is a subgroup of $S_n$. Then without the loss of generality, we can set $\sigma_1 = e$ and $a_1$ to be a vector of ones, and all the other $a_i$, $i > 1$, to be vectors of zeros. This leads to $T(x|a_1, \ldots, a_m) = x$, and thus $T(\cdot|a_1, \ldots, a_m) = I$ when represented as a matrix.

Second, we show that such transforms are closed with binary operation $T^{(1)} \circ T^{(2)}$ defined as $[T^{(1)} \circ T^{(2)}](x) = T^{(1)}[T^{(2)}(x)]$. Specifically, we have

$$[T^{(1)} \circ T^{(2)}](x) = \sum_{i=1}^{m} a_i^{(1)} \odot \sigma_i^{(1)}\left(\sum_{j=1}^{m} a_j^{(2)} \odot \sigma_j^{(2)}(x)\right)$$

$$= \sum_{i=1}^{m}\sum_{j=1}^{m}[a_i^{(1)} \odot \sigma_i^{(1)}(a_j^{(2)})] \odot \sigma_i^{(1)}(\sigma_j^{(2)}(x))$$

Since $K$ is a subgroup of $S_n$, $\sigma_i^{(1)}(\sigma_j^2(\cdot))$ still belongs to $K$. Hence, $T^{(1)} \circ T^{(2)}$ will have the same form as $T(\cdot|a_1, \ldots, a_m)$ after merging like terms.

By representing $T(\cdot|a_1, \ldots, a_m)$ as a matrix, it is clear that the associativity property, i.e., $(T^{(1)} \circ T^{(2)}) \circ T^{(3)} = T^{(1)} \circ (T^{(2)} \circ T^{(3)})$, holds since matrix multiplication is associative. Lastly, the inverse of $T(\cdot|a_1, \ldots, a_m)$ exists since we assume that $T(\cdot|a_1, \ldots, a_m)$ is bijective, and thus its representation is an invertible matrix. $\square$

We want to point out that not all simple matrix-free preconditions can be constructed by Theorem 3.1. Let us take the widely used feature normalization transform, e.g., batch normalization Ioffe and Szegedy (2015), layer normalization Ba et al. (2016) and group normalization Wu and He (2018) as an example. We have

$$T(x|\mu, \sigma) = (x - \mu)/\sigma$$

where $\mu$ and $\sigma$ are either scalar or vector mean and standard deviation of $x$, respectively. This $T(\cdot|\mu, \sigma)$ forms a sparse affine group for $\sigma \neq 0$ Li (2019). However, we cannot use such preconditioners as black-box ones.

## C    CONSTRUCTION OF LOW-RANK APPROXIMATION PRECONDITIONERS

The following property states that the proposed low-rank approximation preconditioners can fit both ends of the spectra of Hessian. It is not difficult to prove this statement. But, it reveals an important advantage over traditional low-rank approximation preconditions with form $P = \rho I + UU^T$, whose eigenvalues are lower bounded by $\rho$.

### C.1    NOTATIONS

Let $Q = (I + UV^T)B$, where $U$ and $V$ are two tall thin matrices, and $B$ is a matrix on certain group, e.g., the group of diagonal matrix, or simply a scalar. It is an important preconditioner as after reparameterization, we can have $Q = \text{diag}(d) + UV^T$ for diagonal matrix $B$, which relates to the LM-BFGS, HF, and conjugate gradient (CG) methods. This preconditioner is efficient if low-rank modication can significantly further reduce the condition number [1] after preconditioning the Hessian with a diagonal matrix, i.e., a Jacobi preconditioner. One also can think $Q$ as a low-rank approximation of the inverse square root of a positive definite Hessian. Note that this form of preconditioner can fit both tails of the spectra of Hessian.

**Claim 3.2.** *Preconditioner $P = Q^T Q$ with $Q = \rho(I + UV^T)$ can have positive eigenvalues arbitrarily larger than $\rho^2$ and arbitrarily smaller than $\rho^2$ with proper $U$ and $V$.*

*Proof.* Let us check the simplest case, i.e., $\rho = 1$, $U = u$ and $V = v$. Then, $P$ is shown to be

$$\begin{aligned} P =& (I + vu^T)(1 + uv^T) \\ =& I + vu^T + uv^T + (u^T u)vv^T \end{aligned}$$

This $P$ has two eigenvalues determined by $u$ and $v$, say $\lambda_1$ and $\lambda_2$. They satisfy

$$\begin{aligned} \lambda_1 \lambda_2 =& (1 + u^T v)^2 \\ \lambda_1 + \lambda_2 =& 2 + 2u^T v + \|u\|^2 \|v\|^2 \end{aligned}$$

By choosing $u$ and $v$ properly, these two eigenvalues can be arbitrarily smaller or larger than 1. For example, by letting $u^T v = 0$ and $\|u\| = \|v\| \to \infty$, we have $\lambda_1 \lambda_2 = 1$ and $\lambda_1 + \lambda_2 \to \infty$. Hence, we must have one eigenvalue arbitrarily large, and the other one arbitrarily small. In general, the order of rank can be larger than 1, and thus more degree of freedoms for fitting the spectra of Hessian. $\square$

---

[1]Since $PH$ is not symmetric, smaller eigenvalue spread does not always suggest smaller condition number, e.g., $[1, a; 0, 1]$ has arbitrarily large conditioner number for $|a| \to \infty$. In PSGD, $P$ does not amplify the gradient noise (ref Li (2015), page 5-6, section IV.B), and thus avoids such degraded solution.

**Claim 3.3.** *If $\rho \neq 0$ and $(I + V^T U)^{-1}$ or $(I + U^T V)^{-1}$ exists, $A_V(\rho, U) = \rho(I + UV^T)$ defines a subgroup of $GL(n, \mathbb{R})$ parameterized with $\rho$ and $U$. Similarly, $A_U(\rho, V) = \rho(I + UV^T)$ defines another subgroup of $GL(n, \mathbb{R})$ parameterized with $\rho$ and $V$.*

*Proof.* Without the loss of generality, we assume $\rho = 1$, and simplify rewrite $A_V(1, U)$ as $A_V(U) = I + UV^T$. We can show that $A_V(U)$ forms a Lie group by revealing the following facts

$$A_V(0) = I$$
$$A_V(U_1) A_V(U_2) = A_V(U_1 + U_2 + U_1 V^T U_2)$$
$$A_V^{-1}(U) = A_V[-U(I + V^T U)^{-1}]$$

i.e., the existence of identity element, closed with respect to matrix multiplication, and the existence of inverse, respectively. The last equation is simply the rewriting of the Woodbury matrix identity, i.e.,

$$[I + UV^T]^{-1} = I - U(I + V^T U)^{-1} V^T$$

The condition that $(I + V^T U)^{-1}$ exists is necessary as otherwise $A_V(U)$ is singular. Lastly, the associativity property clearly holds for matrix multiplications. Hence, all such matrices form a Lie group. Similarly, we can show that $A_U(V) = I + UV^T$ defines another Lie group. $\square$

### C.1.1 THE ROTATION AMBIGUITY AND SCHUR DECOMPOSITION

Note that $UV^T = UQ(VQ)^T$ for any orthogonal matrix $Q$. Thus, we can remove this rotation ambiguity by selecting a $Q$ such that $Q^T(V^T U)Q$ is an upper triangular block matrix with block size 1 or 2, i.e., the Schur decomposition. $A_V(U)$ and $A_U(V)$ still form groups by constraining $V^T U$ to be quasi-triangular.

## D  LOW-RANK APPROXIMATION PRECONDITIONER FITTING

In practice, we seldom use $Q = \rho(I + UV^T)$ as a preconditioner directly. Its limitation is clear, i.e., most of its eigenvalues are $\rho^2$ with $r \ll n$. In our method, we start from a rough preconditioner guess, say $B$, and modify it with $I + UV^T$ to have the refined preconditioner as

$$Q = (I + UV^T)B$$

If matrix $B$ is from another Lie group, we still can update $Q$ efficiently. For example, $B$ can be a diagonal matrix with nonzero diagonals. Then this composited preconditioner reduce to a diagonal one when $r = 0$.

**Computation**  Note that neither of or methods require direct formation of a curvature matrix. Instead we use Hessian vector products. One can utilize auto-differentiation packages to calculate the exact Hessian vector product or approximate them with finite differences both detailed by (Pearlmutter, 1994). Given a neural network with N parameters, the Hessian vector calculation can be done in O(N) time and space, and does not make any approximation. Additionally, we often only calculate the preconditioner with probability p = 0.1, making the PSGD as practical as SGD.

### D.1  FUNDAMENTAL OPERATIONS ON LIE GROUP

### D.1.1  THE CONCEPT OF GROUP GENERATOR

Unlike the additive update to move a point in a Euclidean space, we use multiplicative update to move a point on the Lie group. For example, we can move $A_V(U)$ to any its neighbor, say $A_V(U + \mathcal{E})$, as

$$A_V\left(\mathcal{E}(I + V^T U)^{-1}\right) A_V(U) = A_V(U + \mathcal{E})$$

Since $A_V(\mu U) = I + \mu UV^T = e^{\mu UV^T}$ for $\mu \to 0$, $UV^T$ is a group generator for any $U \neq 0$. Indeed, the Lie algebra is closed as shown by

$$< U_1 V^T, U_2 V^T > = U_1 V^T U_2 V^T - U_2 V^T U_1 V^T = (U_1 V^T U_2 - U_2 V^T U_1) V^T$$

where $< \cdot, \cdot >$ is the Lie bracket.

### D.1.2 THE GRADIENTS FOR PRECONDITIONER UPDATING

Here, the gradient always refers to the one on the Lie group. Thus $dA$, say on group $A_V(U)$, is either $A_V(\mathcal{E})A_V(U)$ or $A_V(U)A_V(\mathcal{E})$, where $\mathcal{E} \to 0$. Since we will update both groups, we simple put it as $dA = \mathcal{E}_1 A$ or $dA = A\mathcal{E}_2$. Let's drop the $E$ in the PSGD preconditioner fitting criterion and derive the stochastic gradient as below,

$$
\begin{aligned}
&d(h^T P h + v^T P^{-1} v) \\
=&h^T dP h - v^T P^{-1} dP P^{-1} v \\
=&2h^T dQ^T Q h - 2v^T P^{-1} dQ^T Q P^{-1} v
\end{aligned}
$$

Since we are to fit $A$ and $B$ on the Lie groups, thus let

$$
\begin{aligned}
dQ =&dAB + AdB \\
=&\mathcal{E}_1 AB + AB\mathcal{E}_2 \\
=&\mathcal{E}_1 Q + Q\mathcal{E}_2
\end{aligned}
$$

Then, we have

$$
\begin{aligned}
&d(h^T P h + v^T P^{-1} v) \\
=&2h^T dQ^T Q h - 2v^T P^{-1} dQ^T Q P^{-1} v \\
=&2h^T Q^T \mathcal{E}_1^T Q h + 2h^T \mathcal{E}_2^T Q^T Q h - 2v^T P^{-1} Q^T \mathcal{E}_1^T Q P^{-1} v - 2v^T P^{-1} \mathcal{E}_2^T Q^T Q P^{-1} v \\
=&2h^T Q^T \mathcal{E}_1^T Q h + 2h^T \mathcal{E}_2^T P h - 2v^T Q^{-1} \mathcal{E}_1^T Q^{-T} v - 2v^T P^{-1} \mathcal{E}_2^T v \\
=&2\mathrm{tr}\left\{\mathcal{E}_1^T \left[(Qh)(Qh)^T - (Q^{-T}v)(Q^{-T}v)^T\right]\right\} + 2\mathrm{tr}\left\{\mathcal{E}_2^T \left[(Ph)h^T - v(P^{-1}v)^T\right]\right\} \quad (19)
\end{aligned}
$$

$$(20)$$

#### D.1.2.1 Gradient with respect to $B$

For diagonal matrix, we simply have

$$0.5\nabla_B = \mathrm{diag}[(Ph)h^T - v(P^{-1}v)^T]$$

from (19), and thus update $B$ as

$$B \leftarrow B(I - \mu\nabla_B)$$

where the step size $\mu$ is small enough such that $\mu\|\nabla_B\| < 1$, and $\|\nabla_B\|$ is just the max absolute diagonal element for a diagonal matrix. Here, $\|\cdot\|$ denotes spectral norm.

For a diagonal matrix, we also can update its diagonals with element-wise step sizes as the Lie group reduces to the direct sum of a series smaller ones with dimension one. This could lead to faster convergence when the true Hessian is diagonal. Otherwise, we do not observe any significant difference between these two step size selection strategies in our numerical results.

#### D.1.2.2 Gradient with respect to $U$ on group $A_V(U)$

Since

$$dA = (I + \mathcal{E}V^T)(I + UV^T) - (I + UV^T) = \mathcal{E}V^T A$$

we replace the $\mathcal{E}_1$ in (19) with $\mathcal{E}V^T$ to obtain gradient

$$0.5\nabla_U = [(Qh)(Qh)^T - (Q^{-T}v)(Q^{-T}v)^T]V$$

Then, we update $A$ as

$$A \leftarrow A - \mu\nabla_U V^T A$$

which suggests its parameter can be updated as

$$U \leftarrow U - \mu\nabla_U(I + V^T U)$$

since we are operating on the group $A_V(U)$, where the step size is small enough such that $\mu\|\nabla_U V^T\| < 1$. Note that $\nabla_U V^T$ has at most rank two. Hence, its Frobenius norm can be used to bound its spectral norm tightly, as shown by

$$\|\nabla_U V^T\|_F/\sqrt{2} \le \|\nabla_U V^T\| \le \|\nabla_U V^T\|_F$$

**D.1.2.3   Gradient with respect to $V$ on group $A_U(V)$**

As

$$dA = (I + U\mathcal{E}^T)(I + UV^T) - (I + UV^T) = U\mathcal{E}^T A$$

we replace the $\mathcal{E}_1$ in (19) with $U\mathcal{E}^T$ to give gradient

$$0.5\nabla_V = [(Qh)(Qh)^T - (Q^{-T}v)(Q^{-T}v)^T]U$$

Thus, we update $A$ as

$$A \leftarrow A - \mu U\nabla_V^T A$$

which implies that its parameter $V$ should be updated as

$$V \leftarrow V - \mu(I + VU^T)\nabla_V$$

where the step size is small enough such that $\mu\|U\nabla_V^T\| < 1$. Again, $U\nabla_V^T$ has at most rank two, and its Frobenius norm gives a tight enough estimation of its spectral norm.

## E   ALGORITHM

---

**Algorithm 1:** PSGD

---

1: Initialize parameter $\theta$ and its learning rate $0 < \mu_1 < 1$
2: Initialize preconditioner as $Q \propto I$, and its update rate and frequency as $0 < \mu_2 < 1$,
   $0 < p \leq 1$, respectively **for** $iteration = 1, 2, \ldots$ **do**
3:
      **end**
      Sample a stochastic loss $\ell(\theta, z)$
4: Compute stochastic gradient $g = \frac{\partial \ell(\theta,z)}{\partial \theta}$ **if** $u < p$ *with* $u \sim \mathcal{U}(0, 1)$ **then**
5:
      **end**
      Sample vector $v \sim \mathcal{N}(0, I)$
6: Compute Hessian-vector product $h = \frac{\partial (v^T g)}{\partial \theta}$
7: Update preconditioner $Q$ with pair $(v, h)$ and rate $\mu_2$
8:
9: Compute preconditioned gradient $\mathfrak{g} = Q^T Q g$
10: Update parameter as $\theta \leftarrow \theta - \mu_1 \mathfrak{g}$
11: Adjust $(\mu_1, \mu_2, p)$ if needed; stop iteration if met certain criteria
12:

---

A few notes for Algorithm 1. Both learning rates, $\mu_1$ and $\mu_2$, are normalized by design. We should not set either of them larger than 1. When the second order derivative is not supported by a certain automatic differentiation tool, we approximate the Hessian-vector product via finite difference as

$$v \sim \mathcal{N}(0, \varepsilon I), \quad h \approx \frac{\partial \ell(\theta + v, z)}{\partial \theta} - \frac{\partial \ell(\theta, z)}{\partial \theta}$$

where $\varepsilon$ is a small positive number, e.g., the machine precision. We should avoid forming the preconditioner $P$ explicitly as $P = Q^T Q$. Instead, the preconditioned gradient is always calculated as $\mathfrak{g} = Q^T(Qg)$. Algorithm 1 is for PSGD with any preconditioner design. Here, we elaborate its preconditioner update algorithms on the two specific Lie groups proposed in this paper. We are not going to detail the algorithm on the calculation of $\mathfrak{g} = Q^T(Qg)$ as its math is pretty straightforward.

---

**Algorithm 2:** XMat Preconditioner Update

---

1: Prepare inputs $Q = \text{diag}(a) + \text{adiag}(b)$ and pair $(v, h)$
2: Calculate $Qh = a \odot h + b \odot \text{flip}(h)$
3: Calculate $Q^{-T}v = (\text{flip}(a) \odot v - \text{flip}(b) \odot \text{flip}(v)) \oslash (a \odot \text{flip}(a) - b \odot \text{flip}(b))$
4: Calculate gradient $\nabla_a = (Qh) \odot (Qh) - (Q^{-T}v) \odot (Q^{-T}v)$
5: Calculate gradient $\nabla_b = (Qh) \odot \text{flip}(Qh) - (Q^{-T}v) \odot \text{flip}(Q^{-T}v)$ **if** $b$ *has odd length* **then**
6:
     **end**
     Set central element of $\nabla_b$ to zero
7:
8: Calculate step size $\mu = \frac{\mu_2}{\max[\max(|\nabla_a|), \max(|\nabla_b|)]}$
9: Update $a$ as $a \leftarrow a - \mu(\nabla_a \odot a + \nabla_b \odot \text{flip}(b))$
10: Update $b$ as $b \leftarrow b - \mu(\nabla_a \odot b + \nabla_b \odot \text{flip}(a))$
11: Return updated preconditioner as $Q = \text{diag}(a) + \text{adiag}(b)$

---

Here are a few notes for Algorithm 2. Notation $\text{adiag}(b)$ denotes the anti-diagonal matrix with elements in vector $b$ as its entries. Notation $\text{flip}(a)$ denotes the operation of flipping the order of elements in vector $a$. For $Q$ with an odd size, we assume that the central element of $b$ is zero to obtain a unique form of representation of the Lie group.

---

**Algorithm 3:** Low Rank Approximation Preconditioner Update

---

1: Prepare inputs $Q = (I + UV^T)\text{diag}(d)$ and pair $(v, h)$
2: Calculate $Qh$
3: Calculate $Ph = Q^T(Qh)$
4: Calculate $Q^{-T}v$ using the Woodbury matrix identity
5: Calculate $P^{-1}v = Q^{-1}(Q^{-T}v)$ using the Woodbury matrix identity
6: Calculate gradient $\nabla_d = (Ph) \odot h - v \odot (P^{-1}v)$
7: Update $d$ as $d \leftarrow d - \frac{\mu_2}{\max(|\nabla_d|)} d \odot \nabla_d$ **if** $u < 0.5$ *with* $u \sim \mathcal{U}(0, 1)$ **then**
8:
     **end**
     Calculate gradient $\nabla_U = (Qh)(Qh)^TV - (Q^{-T}v)(Q^{-T}v)^TV$
9: Update $U$ as $U \leftarrow U - \frac{\mu_2}{\|\nabla_U V^T\|}\nabla_U(I + V^TU)$ **else**
10:
     **end**
     Calculate gradient $\nabla_V = (Qh)(Qh)^TU - (Q^{-T}v)(Q^{-T}v)^TU$
11: Update $V$ as $V \leftarrow V - \frac{\mu_2}{\|U\nabla_V^T\|}(I + VU^T)\nabla_V$
12:
13: Return the updated preconditioner as $Q = (I + UV^T)\text{diag}(d)$

---

Here are a few notes on Algorithm 3. With the Woodbury matrix identity, linear system $Qx = b$ can be solved as

$$x = Q^{-1}b = \text{diag}(1 \oslash d)[b - U(I + V^TU)^{-1}(V^Tb)]$$

which can be separate into the following two steps

$$\text{solve } (I + V^TU)y = V^Tb$$
$$x = \text{diag}(1 \oslash d)(b - Uy)$$

where $I + V^TU$ is a square matrix with size $r$, i.e., the rank or order of low rank approximation. Solving for such a linear system should not be considered as a burden for a moderate $r$, say up to thousands, on today's hardware. Note that $\nabla_U$ is a matrix with rank 2 at most. Thus, we have no need to form $\nabla_U$ explicitly in the actual implementation by writing it as a difference of two outer products,

$$\nabla_U = (Qh)[(Qh)^TV] - (Q^{-T}v)[(Q^{-T}v)^TV]$$

This saves considerable memory and computes for a large $r$. The spectral norm of $\nabla_U V^T$ is approximated with its Frobenius norm. Again, since the rank of $\nabla_U V^T$ is at most 2, relative error of

this approximation is bounded by 3 dB. The same processing techniques apply to the update of $V$ as well. Note that we cannot update both $U$ and $V$ in a single step due to the special way we have constructed these two "twin" Lie groups for low rank approximation. In our implementation, we choose to update either one with equal probability.

In the style of the main here are the algorithms with explicit notation.

**Notations**   we use the following notations:

- $f(\theta) \in \mathbb{R}, \theta \in \mathbb{R}^d$: $f$ is the loss function to minimize, $\theta$ is the parameter in $\mathbb{R}^d$
- $g_t$: the gradient at step $t$
- $p$: preconditioner update probability default is 0.1
- $h_t, v_t$: $h_t$ the Hessian vector product; $v_t$ drawn from $\sim \mathcal{N}(0, I)$
- $P, Q$: $P$ is the gradient preconditioner (never explicitly formed) defined as $P = Q^T Q$
    - $UVd$: For $UVd$, $Q$ takes form of $Q = (I + UV^T)\mathrm{diag}(d)$
    - $X$Mat: For $X$Mat, $Q$ takes form of $Q = \mathrm{diag}(a) + \mathrm{adiag}(b)$
        * $\mathrm{adiag}(b)$: the anti-diagonal matrix with elements in vector $b$ as its entries.
        * $\overline{a}$: the operation of flipping the order of elements in vector $a$.
- $\mu_1, \mu_2$: $\mu_1$ is the optimizer step size; $\mu_2$ is the preconditioner step size, both default to $10^{-2}$

---

**Algorithm 4:** PSGD Optimizer

**Initialize** $\theta_0, t0, Q_0 \propto I$
**While** $\theta_t$ not converged
  $t \; t + 1$
  $g_t \; \nabla_\theta f_t(\theta_{t-1})$
  **If** $u < p$ with $u \sim \mathcal{U}(0, 1)$
    $h_t \; \nabla_\theta (v_t^T g_t)$, s.t. $v_t \sim \mathcal{N}(0, I)$
    **Update** $Q_t$ **via** $(v_t, h_t)$
        Q Update Step
  **Else**
    $Q_t \leftarrow Q_{t-1}$
  $\mathfrak{g}_t \; Q_t^T Q_t g_t$
  $\theta_t \; \theta_{t-1} - \mu_1 \mathfrak{g}_t$

---

**Algorithm 5:** UVd Q Update Step

$Ph = Q^T(Qh)$
$P^{-1}v = Q^{-1}(Q^{-T}v)$     via Woodbury identity 2x
$\nabla_d = (Ph) \odot h - v \odot (P^{-1}v)$
$d \leftarrow d - \mu_2 d \odot \nabla_d / \max(|\nabla_d|)$
**If** $u < 0.5$ with $u \sim \mathcal{U}(0, 1)$
  $\nabla_U = (Qh)(Qh)^T V - (Q^{-T}v)(Q^{-T}v)^T V$
  $U \leftarrow U - \mu_2 \|\nabla_U V^T\|^{-1} \nabla_U (I + V^T U)$
**Else**
  $\nabla_V = (Qh)(Qh)^T U - (Q^{-T}v)(Q^{-T}v)^T U$
  $V \leftarrow V - \mu_2 \|U \nabla_V^T\|^{-1}(I + V U^T)\nabla_V$
**Return** $Q = (I + UV^T)\mathrm{diag}(d)$

---

**Algorithm 6:** XMat Q Update Step

$Q^{-T}v = (\overline{a} \odot v - \overline{b} \odot \overline{v}) \oslash (a \odot \overline{a} - b \odot \overline{b}))$
$\nabla_a = (Qh) \odot (Qh) \text{-} (Q^{-T}v) \odot (Q^{-T}v)$
$\nabla_b = (Qh) \odot \overline{(Qh)} - (Q^{-T}v) \odot \overline{(Q^{-T}v)}$
**If** $b$ has odd length
  Set central element of $\nabla_b$ to zero
$\mu = \frac{\mu_2}{\max[\max(|\nabla_a|), \max(|\nabla_b|)]}$
$a \leftarrow a - \mu(\nabla_a \odot a + \nabla_b \odot \overline{b}))$
$b \leftarrow b - \mu(\nabla_a \odot b + \nabla_b \odot \overline{a})$
**Return** $Q_t \leftarrow \mathrm{diag}(a) + \mathrm{adiag}(b)$

## F MORE EXPERIMENTAL RESULTS

### F.1 NOISY LABEL CIFAR10

SYMMETRIC NOISY LABEL

In this section, we consider training a ResNet18 under $60\%$ symmetric label noise. We randomly select $60\%$ of the CIFAR10 training labels, resulting in each class having approximately $46\%$ correct labels and the $54\%$ consisting of $6\%$ drawn from the other nine classes. Recently, Kwon et al. (2021) proposed a novel flat basin-seeking algorithm named Adaptive Sharpness-Aware Minimization (ASAM) that set a new SoTA for *symmetric noisy label* for CIFAR10.

| Symmetric Noisy Label | Cross Entropy | Smoothened Cross Entropy (0.1) |
|---|---|---|
| PSGD (Ours) | 77.0 | 77.0 |
| ASAM (SoTA) | 70.55 | 70.19 |

We benchmark PSGD against ASAM and see that PSGD outperforms ASAM by $7\%$.

ASYMMETRIC NOISY LABEL

Next, we consider asymmetric label noise. We asymmetrically perturb $60\%$ of the CIFAR10 training labels randomly, resulting in one of the classes having approximately $55\%$ incorrect labels and the other 9 classes having $5\%$ incorrect labels. We use SGD, PSGD, Adam, and Apollo to train a ResNet18 on this task for 100 epochs with 10 different seeds and compare train and test classification accuracy 4. Additionally, we consider the recently proposed flat basin-seeking algorithm named Adaptive Sharpness-Aware Minimization (ASAM) Kwon et al. (2021) that set a new SoTA for *symmetric noisy label* for CIFAR10 under the *asymmetric label noise* setting. We compare ASAM to SGD and PSGD (see Table 5) separately from the other experiments since ASAM is orthogonal yet complementary to SGD and PSGD. To the best of our knowledge, no optimizer has been designed specifically for *asymmetric* label noise.

**Standard Optimization Techniques**

First looking at Figure 4, we see that PSGD achieved the lowest average training accuracy of around $40\%$, with Apollo reaching the highest average train accuracy of $57\%$. While SGD gets an average training accuracy between Adam and Apollo (with 7/8 runs getting $55\%$, and 1/8 getting $34\%$), and well above PSGD. During testing, SGD exhibits clear instabilities, falling into a regime of pure memorization of the training set. This can be seen as among the 8 runs of SGD, only one *lucky initialization* achieved a test accuracy of $77\%$ (learning regime), while the other initializations had a test accuracy of $10\%$ (memorization regime) with an average predicted probability of 0.999. This is not a standard case of over-fitting nor a case of catastrophic forgetting, since the training accuracy does not change. Instead, our experiments show there exists a tipping point in the test accuracy at around epoch 35, where within a single epoch the test accuracy drops from $71\%$ to $10\%$ while the training accuracy shows no intelligible indication of this change. Furthermore, the test accuracy does not increase for the rest of the 65 epochs. At the 35 epoch mark, Adam and Apollo both go through a period of high variance but eventually converge (due to the cosine learning rate decay). PSGD achieved an average accuracy of $82.63\% \pm 0.11$ after 100 epochs, outperforming SGD by $63.93\%$ and exhibiting no optimization instabilities (See Figure 4).

**Sharpness Aware Optimizers**

In this section we compare Adaptive Sharpness-Aware Minimization (ASAM) Kwon et al. (2021) that set a new SoTA for *symmetric noisy label* for CIFAR10 under the *asymmetric label noise* setting to PSGD and SGD. Note to the best of our knowledge there has not been an optimizer specifically designed for asymmetric label noise and no specific optimizer has been deemed SoTA.

We see in Table 5 that while ASAM is able to achieve the best training accuracy we see that greatly overfits the dataset resulting $6\%$ test accuracy at the end of 100 epochs, $12.38\%$ below SGD and $76.31\%$ below PSGD. While ASAM may have been the SoTA for Symmetric label noise, clearly another solution is needed under the Asymmetric setting.

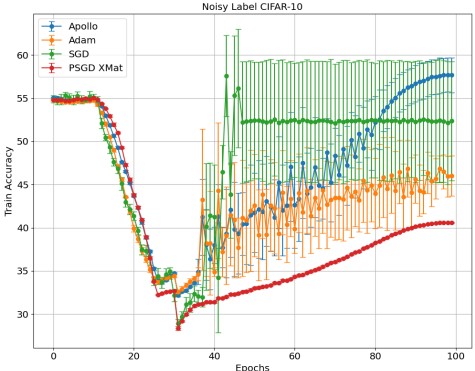 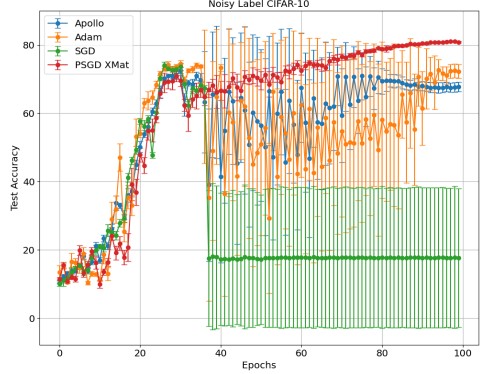

(a) Train Accuracy: Noisy Label CIFAR-10      (b) Test Accuracy: Noisy Label CIFAR-10

Figure 4: CIFAR-10 Noisy Label ResNet-18: (a) Train Acc: We clearly see that Apollo (Quazi-Newton diagonal) over-fits the training set which includes 60% noisy labels, whereas PSGD finds a solution that results in a significantly worse train accuracy. Furthermore, we see that the variance for Apollo, Adam, and SGD is very high during training starting around epoch 40. Note SGD had 7 runs with train accuracy near 55%, and one with train accuracy around 34%. (b) Test Acc: We see that PSGD reaches a significantly better test accuracy with a very low training variance compared to Apollo, Adam, and SGD. In this run, SGD had 7 NNs that got 10% test accuracy and one network that got 72% test accuracy. For more on SGD's performance see main paper **CIFAR with Noisy Lables**. The combination of these two plots shows other optimizers can easily over-fit/memorize incorrect image label pairs, can often have large optimization instabilities during training, and even reach degenerate solutions where the NNs have reasonable train accuracy but random test accuracy.

Table 5: Comparing (P)SGD to Sharpness Aware Optimizers. Here we see that PSGD can outperform SGD and ASAM by 12.38% and 76.31% respectively.

| Asymmetric Noisy Label CIFAR10 | Final | | Best | |
|---|---|---|---|---|
| | Train | Test | Train | Test |
| PSGD (Ours) | 40.44% ± 0.12 | 82.63% ± 0.11 | 41.03% ± 0.11 | 82.63% ± 0.11 |
| SGD | 52.5% ± 19.87 | 18.7% ± 33.75 | 56.3% ± 0.1 | 73.98% ± 5.65 |
| ASAM | 64.03% ± 1.12 | 6.32% ± 2.6 | 64.03% ± 0.1 | 40.48% ± 0.8 |

As such we see that PSGD, which is a general-purpose optimizer, is able to outperform general first and second-order optimizers as well as flat minima-seeking optimizers under the noisy label regime.

### F.2 MNIST HANDWRITING DIGIT RECOGNITION

To have a fair comparison between the diagonal and low-rank approximation (LRA) preconditioners, we slightly upgrade $Q$ in the LRA preconditioner to form

$$Q = \text{diag}(d)(I + UV^T)$$

where $d$ is a vector. This form of $Q$ cannot form a Lie group. Still, its two factors, $\text{diag}(d)$ and $I + UV^T$, can be fitted on their own Lie groups. Now, the diagonal preconditioner is a special case of this LRA one with order $r = 0$.

We have tested orders $r = 0, 1, 2, 5$, and $10$. The batch size is $64$. Totally ten epochs of training are performed. The learning rate for parameter updating is annealed exponentially from $0.1$ for the first epoch to $0.001$ for the tenth epoch. The step size for preconditioner fitting is annealed exponentially from $0.1$ for the first epoch to $0.01$ for the tenth epoch. The preconditioned gradient norm is clipped to $10$ if too large. No momentum is used. Figure 1 summarizes the test classification error rates for preconditioners with different orders of approximations over 50 runs. From Fig. 1, we see that the simple diagonal preconditioner performs well. Still, LRA brings marginal gains up to order $r = 5$. This cost function is fairly 'flat' since the only nonlinearities in LeNet5 are two piece-wise linear functions, i.e., the activation function ReLU and max pooling one. Only the cross entropy loss introduces the 'curvature'.

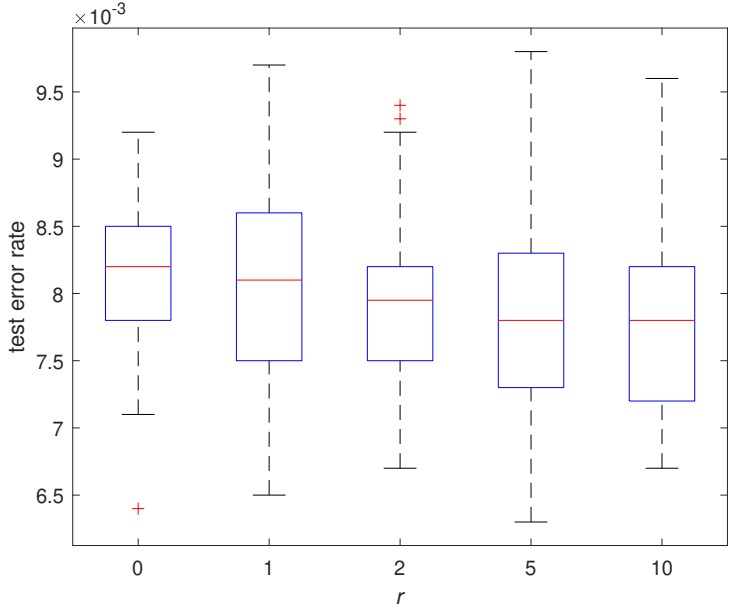

Figure 5: MNIST test classification error rates over 50 runs using preconditioners with different orders of LRA. The one with order 0 reduces to the diagonal preconditioner. Table 1 reports results of classification accuracy for $r = 5$. Higher is better.

### F.3 TOY EXAMPLE: INVESTIGATING THE FLATNESS OF SAM BASED SOLUTION

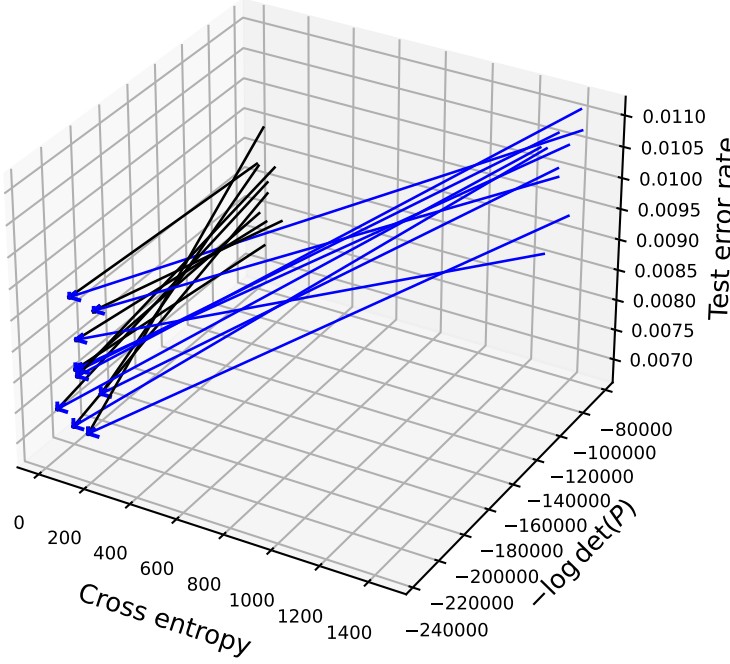

Figure 6: (Adam,ASAM)→ PSGD: MNIST hand written digit recognition with LeNet5. Hessians at the minima of Adam are estimated with a dummy LRA PSGD optimizer that only updates the preconditioner.

Very recently sharpness-aware minimization (SAM) based optimizers have become a popular area of research. We were interested to see if PSGD compares to SAM in a toy MNIST LeNet5 classification task. We discussed in 5.1, how the smaller the $-logdet(P)$ of a NN is, the flatter the solution found by the optimizers. Since SAM has been specifically designed to find flat optima we would like to

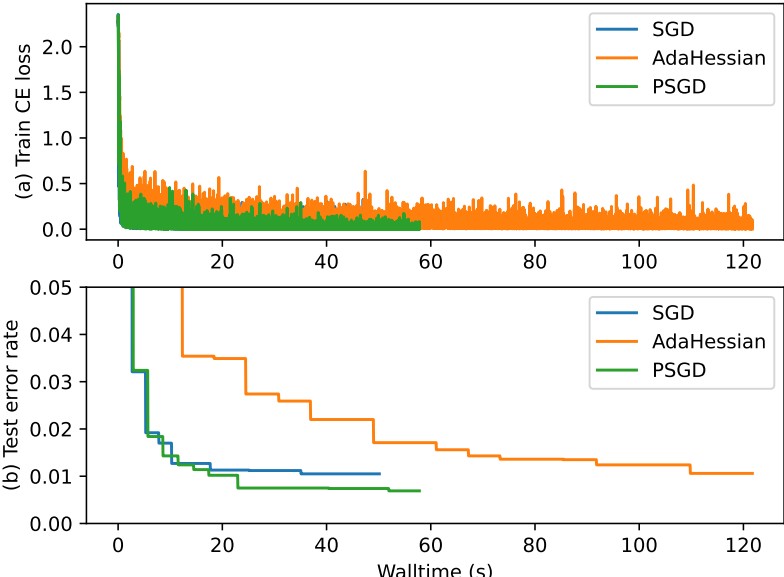

Figure 7: Wall-Time: LeNet5 Comparing SGD, PSGD, and AdaHessian. We see that PSGD UVd (rank 10 Hessian estimate) takes 1.2x the time of SGD but finds a better solution. AdaHessian (Hessian diagonal estimate) takes 2x to complete the run and reaches an error rate compared to both SGD and PSGD.

compare. Fig. 6 shows ten pairs of minima, each starting from the same random initial initialization. We see that PSGD converges to minima with flatter or smaller Hessian, i.e., larger preconditioners. From the view of information theory, the total cross entropy and $0.5 \log \det(H) \approx -0.5 \log \det(P)$ are good proxies of the description lengths of the train image-label pairs and model parameters, respectively. Fig. 6 shows that minima with smaller description lengths tend to perform better on the test sets as well, suggested by an arrow pointing to the down-left-front corner of the cube.

This shows that we can actually find flatter minima compared to SAM-based optimizers without explicitly optimizing for them.

### WALL-CLOCK TIMINGS

Theoretically, one Hessian-vector evaluation doubles the complexity of one gradient evaluation, since we do this with a probability of 0.1, the per iteration complexity of PSGD becomes (1+0.1*2) = 1.2 times that of SGD. Empirical timings for LeNet5 added in the general response (Figure 7) align with the theoretical calculation. We see that PSGD LRA (Low-Rank Hessian Approximation) with a rank of 10's overhead is minimal compared to SGD and is 2x faster than AdaHessian which uses a diagonal Hessian estimate.

### DYNAMICS OF PRECONDITIONER

The dynamics of the preconditioner depend on the specific network structure and task to study. Suppose we start from a moderate initial guess for the preconditioner (by default it is set to 1). For many tasks, we have observed that the max eigenvalue of the preconditioner keeps increasing until saturation to a point during the whole learning process. As the maximum eigenvalue $P$ of corresponds to the inverse of the minimum eigenvalue of $H$, this is expected for over-parameterized NNs as their minimum eigenvalue approaches 0. The minimum eigenvalue of $P$ first increases during the early stages of learning, and then eventually drops and settles down on a small value, suggesting parameters are locked along those eigenvector directions associated with small eigenvalues. These empirical results coincide with independent findings from many authors showing that network learning has two stages: the early growing one and the latter refining one. We showed the dynamics of the preconditioner on the LeNet5 (see Figure 8). For this specific task, the min eigenvalue occasionally drops but keeps increasing during the last stage of learning due to the vanishing Hessian phenom-

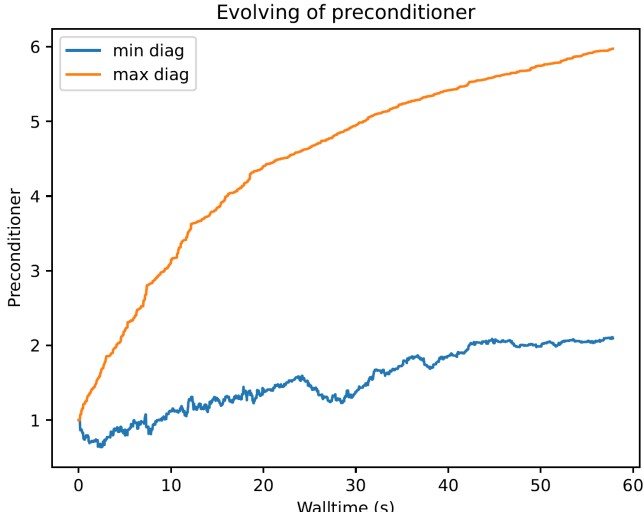

Figure 8: Dynamics of Preconditioner for LeNet5: The max eigenvalue of preconditioner keeps increasing until saturation to a point during the whole learning process. For this specific task, the min eigenvalue also keeps increasing during the last stage of learning due to the vanishing Hessian phenomena when the train cross-entropy loss approaches zero.

ena when the train cross-entropy loss approaches zero (meaning the absolute logistic outputs can be scaled to be arbitrarily large, thus gradient and Hessian all approaching 0 on the train set).

### F.4 CIFAR10 IMAGE CLASSIFICATION WITH RESNET18

We follow the implementations of Adabelief[2] algorithm Zhuang et al. (2020) to test preconditioned SGD (PSGD) on the CIFAR10 image classification task with the ResNet18 model. One main difference from the implementations in Zhuang et al. (2020) is that we reduce the learning rate by tenfold twice for all the optimizers, while the original stage learning rate scheduler only anneals the step size once. We also consider the cosine learning rate scheduler, which helps SGD to achieve the state-of-the-art (SOTA) test accuracy about $95.5\%$. Training and testing accuracy convergence curves over 16 runs are plotted in Fig. **??**. We only show the results of PSGD and SGD here as SGD is known to achieve the SOTA results for this problem.

For PSGD, we use step size $0.02$ for parameter updating and $0.01$ for preconditioner fitting. The preconditioner is only updated once per ten iterations, and thus its overhead over SGD is marginal. The same momentum factor, $0.9$, is used for both SGD and PSGD. Since the step size in PSGD is normalized, we update the momentum as $m \leftarrow 0.9m + 0.1g$, instead of $m \leftarrow 0.9m + g$ as in the SGD. No gradient clipping is used. Weight decay is realized by adding the L2 regularization term $0.5\lambda\theta^T\theta$ to the cross entropy loss. We have found that $\lambda$ between $0.01$ and $0.02$ performs the best.

From Fig. **??**, we observe that SGD performs very well with the cosine learning rate scheduler. This is expected as these residual networks are highly evolved to be first-order optimizer-friendly. The extensive use of piece-wise linear functions, residual connections, and batch normalizations make these models fairly 'flat' and resemble shallow models, instead of deep ones Veit et al. (2016). Still, PSGD slightly outperforms SGD when we remove the shortcut connections or use a less-tuned learning rate scheduler, e.g., stage one here.

#### PSGD IS ROBUST TO HYPER-PARAMETERS

To showcase the robustness of PSGD to hyper-parameters we show in table 6 the test classification accuracy of a ResNet18 trained on CIFAR10 using different learning rates and weight decay.

We clearly see the test accuracy of PSGD in a wide range of learning rates and weight decay converges within the expected range of $95.49 \pm 0.08\%$.

---

[2]https://github.com/juntang-zhuang/Adabelief-Optimizer

| PSGD | Learning Rate | Weight Decay | Test Accuracy |
|------|---------------|--------------|---------------|
| XMat | 2e-2 | 2e-2 | 95.32 |
| LRA | 2e-2 | 2e-2 | 95.69 |
| LRA | 5e-2 | 2e-2 | 95.48 |
| LRA | 5e-2 | 2e-2 | 95.46 |
| LRA | 4e-2 | 2e-2 | 95.55 |
| LRA | 3e-2 | 2e-2 | 95.57 |
| LRA | 2e-2 | 1e-2 | 95.45 |
| LRA | 2e-2 | 3e-2 | 95.51 |

Table 6: We see that PSGD is robust to hyper-parameter selection making tuning significantly easier compared to other optimization methods.

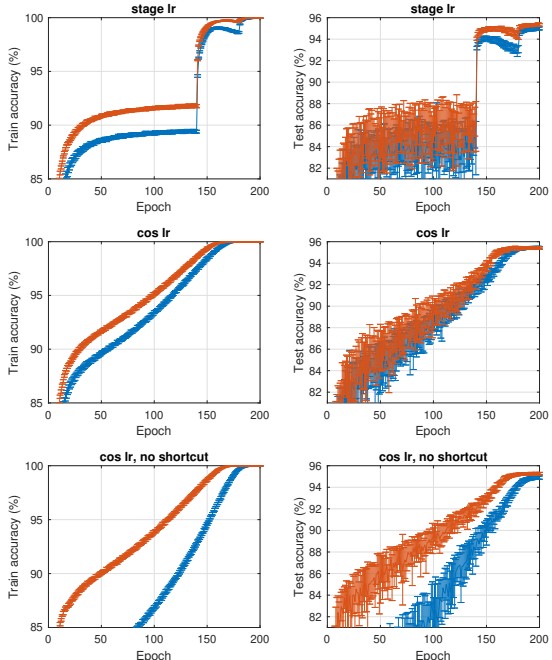

Figure 9: CIFAR10 image classification with ResNet18. The order of low-rank Hessian approximation is 10. Mean and variance are estimated over 16 runs. Higher is better.

## F.5 A LARGE-SCALE LOGISTIC REGRESSION PROBLEM

We consider a simple logistic regression model to solve the MNIST classification problem, with and without Bernoulli noise. We considered AdaHessian, AdaBelief, SGD, LBFGS, PSGD LRA/XMat, KFAC, and HessianFree optimizers. Let $x$ be the vector of the image with length $28^2$. Instead of regression on vector $x$, we do the regression on the outer product vector of $x$, which has length $28^4$. This significantly increases the test classification accuracy but leads to a large regression matrix with over six million coefficients. The KFAC preconditioner did not fit on our GPU and the HessianFree optimizer would diverge far from all other optimizers in more than 50% of our runs. Both are omitted from further discussion in this section.

No momentum is considered since this is the case for LM-BFGS. The train batch size is $500$. It is tricky to select the initial learning rate for LM-BFGS even though we exponentially anneal it. We have found that LM-BFGS diverges on roughly one-third of the trials with an initial learning rate of $0.1$, but $0.05$ is too small and may lead to worse performance than SGD. For PSGD, we consider the LRA preconditioner with order $10$ and set the learning rates for parameters and preconditioner to $0.05$ and $0.1$, respectively. Since LM-BFGS might diverge with a learning rate of $0.1$, we only show

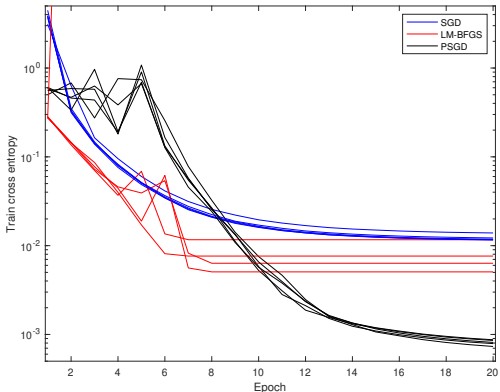

Figure 10: Standard MNIST dataset: Typical convergence curves on the logistic regression problem. Lower is better. When comparing the convergence speed, one should be aware that one step of LM-BFGS may take up to ten iterations, while SGD and PSGD always have one iteration per step.

a few typical convergence curves of SGD, LM-BFGS, and PSGD in Fig. 10. LM-BFGS converges to regression losses a few times smaller than SGD. PSGD could converge to losses about one order of magnitude lower than that of SGD and LM-BFGS. Regarding test classification error rate, we have $2.37\% \pm 0.08$, $2.09\% \pm 0.18$, $1.98\% \pm 0.08$ for SGD, LM-BFGS, and PSGD, respectively, averaged over ten runs. Again, LM-BFGS outperforms SGD, and PSGD performs the best on the test classification error rate as well.

Next, we simply randomly add Bernoulli noise to the MNIST dataset to add diversity to the dataset. Even though LM-BFGS is the standard optimizer for large-scale logistic regression we wanted to consider some other SOTA optimizers for deep learning. PSGD UVd/XMat performed the best in this scenario but we found that AdaBelief did surprisingly well given its lack of second-order information. Losses and Accuracies plotted in Fig 11

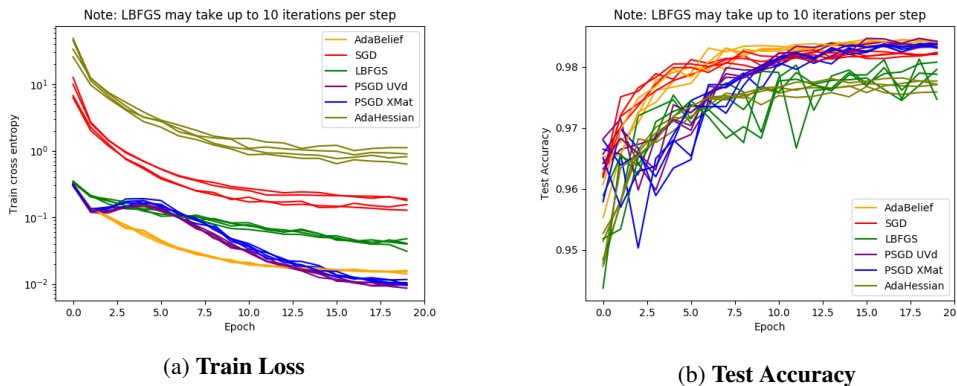

(a) **Train Loss**                                    (b) **Test Accuracy**

Figure 11: We consider logistic regression on the MNIST dataset with Bernoulli noise. We see PSGD finds the lowest loss on the train set, with the Belief mechanism also working well. PSGD and AdaBelief generalize well to the Accuracy of the Test Dataset.

We see that PSGD significantly outperforms the SOTA optimizers in the convex logistic regression setting under noise-free or noisy domains while being memory efficient.

### F.6 FORGETTABILITY & UNCERTAINTY: RANK ANALYSIS

Very recently, Toneva et al. (2018) shows that Forgettable points are crucial for generalization in the supervised setting. In the unsupervised, semi-supervised, self-supervised, and generative setting Pooladzandi et al. (2023) shows that one can use the low entropy samples generated by the generative model to significantly boost the classification performance of the Latent Energy Based Model which acts as generative-classifier. Here we acknowledge the previous findings and focus on the supervised

setting. We train a ResNet18 on CIFAR-10 and record entropy and forgettability statistics. We plot the strong correlation between low forgettability score and low entropy found in our statistics in Fig. 12 and provide correlation coefficients in Table 7.

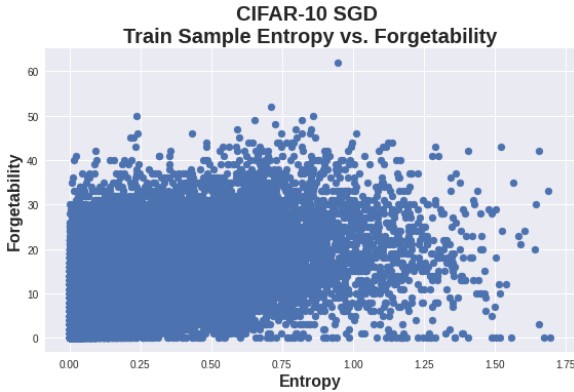

Figure 12: There is a strong correlation between the forgettability ordering and the entropy ordering. i.e. unforgettable points have a very low entropy.

| Entropy v Forgettability Score | Correlation Coefficient | | p-value | |
| --- | --- | --- | --- | --- |
| | SGD | PSGD | SGD | PSGD |
| Spearman | 0.72 | 0.73 | 0 | 0 |
| Pearson | 0.57 | 0.65 | 0 | 0 |
| Kedal Tau | 0.54 | 0.56 | 0 | 0 |
| Weighted Tau | 0.60 | 0.75 | 0 | 0 |

Table 7: Comparison of correlation metrics comparing Entropy Score vs Forgettability score between SGD and PSGD. We see that PSGD's average correlation between entropy and forgettability is stronger than that of SGD.

### GENERALIZATION GAP BETWEEN HIGH AND LOW ENTROPY SUBSETS

We train an Oracle LeNet5 on the full MNIST dataset for 20 epochs, we categorize the train set into two subsets; a high entropy subset and a low entropy subset, each consisting of 10k points. The entropy is defined over the softmax of the logits. We then train two different LeNet5s on each dataset. We find that the low entropy dataset does not generalize to the test set well achieving a test accuracy of 74%, whereas the high entropy dataset achieves a 99.3% which is on par with Oracle LeNet5 (see 8). This clearly shows for supervised learning the high entropy points are most important for generalization.

Furthermore, we find that training on low entropy points results in low entropy and a lower mean entropy network. This supports the hypothesis that there are certain points that the net can lower the entropy over which do not lead to generalization. We see in Table 8 that training over the full dataset resulted in a higher mean low entropy compared to that of the full dataset. This supports the idea that high entropy points are important for supervised learning.

In terms of distance from initialization, we see that the network is pushed farther from initialization when using the full dataset than while using the high entropy points and the least when using the low entropy points. This supports that training with low-entropy or unforgettable points unnecessarily pushes a network's parameters far from initialization reducing generalization capacity. Furthermore, we see that when training on low entropy points, PSGD does not push the neural network far from initialization preserving generalization capacity Cao and Gu (2019).

Finally, we see that an NN trained on low entropy points has a mean low entropy 2 orders of magnitude smaller than training on the full dataset, and 8 orders of magnitude smaller than training on the high entropy subset. This shows that low entropy points cause a level of confidence in a NN which is not needed and in some cases can be hurtful to generalization 5.2.

| MNIST Statistics | Accuracy | Expected Low Entropy | Expected High Entropy | Distance from Initilization SGD | Distance from Initialization PSGD |
|---|---|---|---|---|---|
| Full Dataset | 99.3% | 5e-16 | 6e-3 | 23.1 | 26 |
| High Entropy Subset | 99.3% | 1e-10 | 2e-2 | 21.7 | 21.2 |
| Low Entropy Subset | 74.2% | 6e-18 | 4e-4 | 9.2 | 8.7 |

Table 8: Comparing generalization accuracy of a LeNet5 trained on either 10k high or 10k low entropy data-points. We see high entropy datasets can match generalization of the full dataset whith a higher test entropy compared to the other datasets. We see that the distance from initialization is less for the low entropy points.

### F.7 EFFECT OF RANK ON XOR

The full rank version of PSGD Li (2019) can solve the XOR problem with an RNN past delay of 128. Since we can arbitrarily increase the rank of Hessian approximation in our LRA version of PSGD, we consider the effect of rank on convergence on the XOR problem using an RNN. We find rank approximations of $r = 0, 1, 2, 5,$ and 10 converge with probability $p = 0.1, 0.4, 0.8, 1$ and 1 respectively.

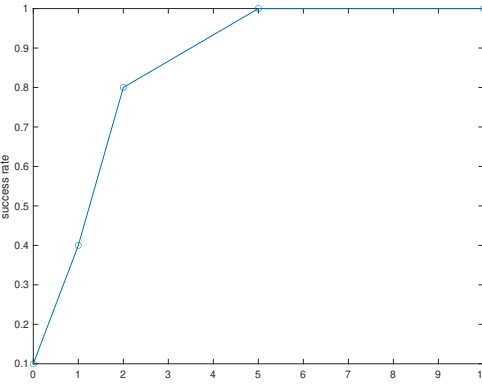

Figure 13: Success rate over ten runs in solving the XOR problem with a simple RNN and LRA preconditioners of orders 0, 1, 2, 5, and 10. Higher is better.

This example shows a typical problem where the diagonal preconditioner struggles, while a low-order Hessian approximation works perfectly for preconditioning.

For all experiments, both step sizes for parameter and preconditioner updating are fixed at $0.01$. The gradient clipping threshold is set to $1$. No momentum is used. We run 10 runs per optimizer for $100,000$ iterations or until convergence. The success rates over ten runs for each r are plotted in Fig. 13. This example shows a typical problem where the diagonal preconditioner struggles, while a low-order Hessian approximation works perfectly for preconditioning.

SETTING A BASELINE USING KFAC, $PSGD_A$ $PSGD_{LRA/XMat}$, SGD AND ADAM

In this section, we compare general-purpose black box versions of PSGD, namely $PSGD_{LRA}$ and $PSGD_{XMat}$ to the Affine or Kronecker Factorized version of $PSGD_A$, to KFAC, SGD, and Adam. While $PSGD_A$ is able to outperform the other optimizers, the Affine version of PSGD requires careful adjustment of neural network architecture to be used. This becomes infeasible for large and intricate modern NN architectures. We see in Table 9 that the black box variants nearly match the test accuracy of $PSGD_A$ outperforming the memory-hungry KFAC (second order) optimizer, as well as first order SGD and Adam.

**Removing Skip Connections** To increase curvature in the relatively flat modern ResNet architecture, we remove the residual skip connections that gave ResNets their namesake. A recent study Zhang et al. (2022) demonstrated the use of Tailored Activation Transformation (TAT) in conjunc-

Table 9: Test accuracy of LeNet5 on MNIST over 10 runs.

| SGD | Adam | KFAC | $PSGD_A$ | $PSGD_{XMat}$ | $PSGD_{LRA}$ |
|---|---|---|---|---|---|
| 99.04 | 99.12 | 99.16 | 99.26 | 99.22 | 99.22 |

Table 10: Stage & cosine learning rate and residual-free ResNet18-RF on CIFAR10.

| | ResNet18 | | ResNet18-RF | |
|---|---|---|---|---|
| $lr$ | *cos* | *stage* | *cos* | *stage* |
| SGD | 95.51 | 95.07 | 94.97 | 94.35 |
| PSGD | 95.54 | 95.43 | 95.36 | 95.17 |

Table 11: Test Accuracy of RN18 on diverse CIFAR10 derived tasks with (P)SGD, Adam & Apollo (2nd order). The current SoTA optimizer is *italicized*, and the best accuracies are bolded.

| CIFAR10 | Standard | No Shortcut | Class Imb | Noisy Final | Noisy Best | Adversarial | Blur Deficit |
|---|---|---|---|---|---|---|---|
| PSGD (Ours) | **95.49**$_{0.08}$ (5.5hrs) | **95.27**$_{0.09}$ | **87.16**$_{0.85}$ | **82.63**$_{0.11}$ | **82.63**$_{0.11}$ | **85.17**$_{0.04}$ | **89.51**$_{0.12}$ |
| SGD | $95.47_{0.14}$ **(4.5hrs)** | $94.66_{0.16}$ | $86.32_{0.84}$ | $18.7_{33.75}$ | $73.975_{5.65}$ | $83.82_{0.18}$ | $83.51_{0.15}$ |
| Adam | $93.15_{0.02}$ **(4.5hrs)** | $92.70_{0.02}$ | $83.18_{1.2}$ | $72.47_{2.17}$ | $74.47_{1.06}$ | $82.26_{0.49}$ | $82.62_{0.015}$ |
| Apollo | $90.59_{0.252}$ (5hrs) | $92.00_{0.20}$ | $78.23_{1.3}$ | $67.71_{1.49}$ | $72.58_{0.92}$ | $73.9_{0.57}$ | $79.25_{0.46}$ |

tion with K-FAC Martens and Grosse (2015) (another second-order optimizer) to close the gap between residual networks with and without residual skip connections. However, in our experiment, we do not utilize TAT and instead compare the performance of SOTA optimizers on both residual-free and standard ResNet18 models. The results, summarized in Table 10, indicate that PSGD outperforms SGD by $0.61\%$ and $0.20\%$ on residual-free and standard ResNet18, respectively. Our findings are consistent with the results from Zhang et al. (2022), where a difference of $0.6\%$ was observed between the optimization of residual-free using TAT and standard ResNet18.

**Class Imbalanced CIFAR10** We evaluate the optimization performance on a class-imbalanced version of the CIFAR10 dataset, where $50\%$ of the classes are randomly reduced by an order of magnitude. We compare SGD and PSGD on optimizing ResNet18 and report the results in Table 11. Our results show that PSGD outperforms the SOTA by $1.03\%$ on this dataset.

**Adversarial Attacked CIFAR10** Finally, we trained a ResNet18 on 10k unmodified CIFAR10 images and evaluated it on a test set consisting of 40k samples perturbed using Neural Tangent Generalization Attacks Yuan and Wu (2021). As shown in Table 11, PSGD outperformed SGD by $1.35\%$.

**Forgettability Statistics and Learning** We revisit Toneva et al. (2018)'s forgettability experiments, which found that one can prune $30\%$ of the CIFAR-10 train samples without loss of generalization. The study found that a point's utility for generalization increases as it is learned and then subsequently forgotten during training, regardless of the optimizer or architecture used. Essentially, forgettability ordering shows which data points an NN uses to define high-dimensional boundaries akin to support vectors. We investigate whether the performance difference between PSGD and SGD's generalization performance can be attributed to this forgettability ordering.

We train the RN18 four times, keeping the top $N$ important points based on each optimizer's expected forgettability score. Table 13 shows that PSGD focuses on points that are central to generalization. We see this since, when we limit the dataset to only the 5k most forgettable data points deemed by each optimizer, we see PSGD is able to outperform SGD by nearly $14\%$.

SGD and PSGD exhibit fundamentally different importance orderings, which is evident from the significant generalization gap observed when training on pruned datasets using these orderings at various degrees. We find that PSGD and SGD have a statistically significant correlation between their forgettability orderings, but the strength of the correlation is weak (Spearman coefficient of 0.02 and a p-value of $p = 1x10^{-12}$). This indicates that the nature of training observed through forgettability is different for PSGD compared to first-order optimizers.

Table 12: Uncertainty Statistics: PSGD's higher uncertainty leads to better generalization and less over-fitting.

| NTK | Entropy | | | Margin | | |
|---|---|---|---|---|---|---|
| | Min | Mean | Max | Min | Mean | Max |
| PSGD | 0.139 | 0.260 | 1.545 | 0.144 | 0.956 | 0.994 |
| SGD | $7x10^{-7}$ | 0.01 | 0.8975 | 0.3925 | 0.999 | 1 |
| Adam | $1.5x10^{-7}$ | 0.009 | 0.8645 | 0.3625 | 0.999 | 1 |
| Apollo | $1x10^{-6}$ | 0.05 | 0.8851 | 0.4326 | 0.999 | 1 |

Table 13: Forgetting statistics for CIFAR10 on ResNet18. PSGD finds better forgettability statistics outperforming SGD.

| Forgetting | 50k | 25k | 15k | 5k |
|---|---|---|---|---|
| PSGD | 96.65 | 95.83 | 94.95 | 56.46 |
| SGD | 96.21 | 95.56 | 93.7 | 42.48 |

Furthermore, we see a strong correlation coefficient of 0.75 and 0.60 between a low forgetability score and low entropy score with a p-value of $p = 0$ for PSGD and SGD respectively (see Fig 12). Hence, PSGD does a better job of shaping the NN to focus on the highly forgettable points that are important for generalization while not becoming overconfident on the easy unforgettable points giving up too much capacity, unnecessarily pushing the parameters away from initialization reducing generalization ability Cao and Gu (2019) (see 8). Note while Toneva et al. (2018) shows that forgettable points are more important for generalization for supervised learning, very recently Pooladzandi et al. (2023) showed low entropy points are the more important points for learning in the unsupervised, semi-supervised, and generative model setting. See Table 8 showing generalization gap training low and high entropy points and how it affects the distance from initialization.

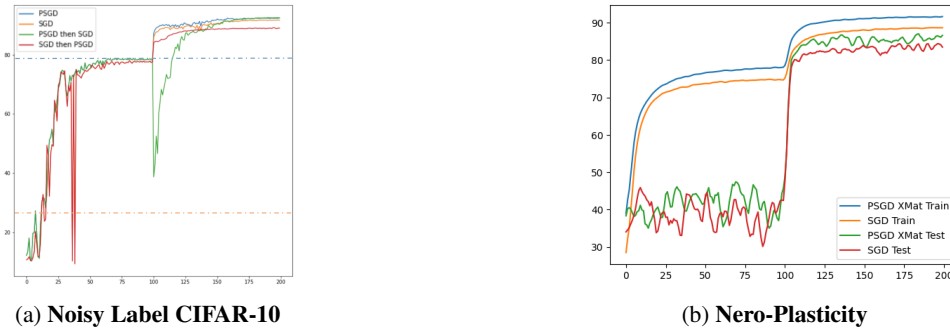

(a) **Noisy Label CIFAR-10**  (b) **Nero-Plasticity**

Figure 14: a) Solutions found by SGD are harder to optimize compared to PSGD. Note for SGD we used a lucky initialization (see 5.2). The blue and yellow dotted line are the average accuracy of PSGD and SGD after 100 epochs respectivly. b) Removing the blur-deficit at epoch 100, PSGD is be more neuro-plastic compared to SGD, achieving better train and test performance.

**Stubborn Solutions of First Order Optimizers** From the previous examples, we learned that first-order optimizers tend to heavily overfit certain data points, reducing entropy. In the case of noisy labels, this often results in pure memorization where the network achieves the training accuracy of PSGD on the train set, but only 10% accuracy on the test dataset with an average confidence of 99.9%. To better understand the stubbornness of SGD solutions, we consider the *lucky initialization,* which resulted in a test accuracy of 77% under noisy label conditions. Here, we examine a transfer learning problem, where a ResNet18 was trained on noisy data for 100 epochs and then on clean labels for another 100 epochs. This simulates a real-world distributional shift where noisy labels may be refined throughout training, leading to a distributional shift. We compare neural networks trained with each optimizer, as well as those that changed optimizers after 100 epochs of training.

As seen in Fig 14a, PSGD finds a flexible solution when given noisy labels. This is seen since when we correct the labels both PSGD and SGD can reach an accuracy of 92.42%. In contrast, the solution found by SGD seems to be stubborn since when we correct the noisy labels, PSGD then SGD reaches an accuracy of 91.7% and SGD then PSGD has an accuracy of 88%. This shows the importance of curvature information in the early periods of training.

Intuitively, first-order methods cause NNs to dedicate parameters to memorizing some data points, unnecessarily reducing entropy. When memorization occurs given incorrect labels, it may be difficult to reshape the NN when the labels are corrected, leading to the loss in generalization accuracy seen in Fig 14a. In contrast, as PSGD finds a less certain or suborn solution in the first stage of training, either optimizer can then reshape the NN to their liking in the second stage.

We believe the noisy-label and neuro-plasticity results have strong applicability to transfer learning and coreset selection Pooladzandi et al. (2022), particularly in scenarios where the training distribution changes during training. Recently, Osawa et al. (2022) demonstrated that the affine Lie group variant of PSGD outperforms other optimizers when fine-tuning a ResNet18 and ViT Dosovitskiy et al. (2020) on CIFAR10 that were pre-trained on ImageNet Deng et al. (2009).

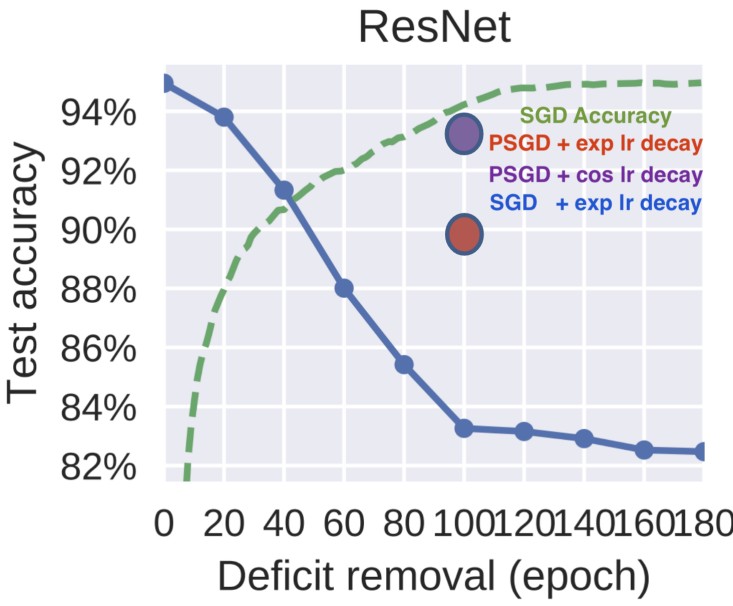

Figure 15: Defecit plot: We see that PSGD is able to close the gap

**More Experiments on Neuro Plasticity**

Neuroplasticity experiments done by Achille et al. (2017) provided an interesting insight into the critical learning periods of NNs. Here we consider whether PSGD can extend the critical learning period of an NN. The summary of Achille et al. (2017) , is that if there is a deficit in learning that is not removed by the first 80 epochs, the final test accuracy will be significantly hindered.

In Figure 15, we see that if we remove the blur at epoch 100, well after the end of the critical learning period of an NN trained with SGD/Adam, PSGD is able to retain classification accuracy as if we removed the deficit around epoch 50. If we switch to a cosine learning rate schedule, PSGD is able to recover the accuracy as if one removed the deficit at 20 epochs.

We believe that this nature of PSGD is due to us finding a flat generalizable solution that is not memorizing points. Since we are not memorizing, and keeping our entropy relatively high compared to other first and second-order optimization methods, we are able to recover accuracy and reshape our NN when the deficit is removed.

Table 14: Test perplexity (lower is better) on Penn Treebank for one-, two- and three-layered LSTMs.

| LSTM | PSGD | Adan | AdaBelief | SGD | AdaBound | Adam | AdamW | Padam | RAdam | Yogi |
|---|---|---|---|---|---|---|---|---|---|---|
| 1 layer | **83.5** | 83.6 | 84.2 | 85.0 | 84.3 | 85.9 | 84.7 | 84.2 | 86.5 | 86.5 |
| 2 layers | **64.9** | 65.2 | 66.3 | 67.4 | 67.5 | 67.3 | 72.8 | 67.2 | 72.3 | 71.3 |
| 3 layers | **59.7** | 59.8 | 61.2 | 63.7 | 63.6 | 64.3 | 69.9 | 63.2 | 70.0 | 67.5 |

**LSTM based models** We conduct a performance comparison between second-order optimization methods and first-order methods for training recurrent neural networks (RNNs) and transformer models. RNNs have a recurrent structure that exhibits strong curvature properties, making them particularly suitable for second-order optimization methods. Such methods can efficiently leverage this structure to converge quickly to good solutions. Consequently, RNNs usually perform better when trained with second-order optimization methods Martens (2016), particularly when the objective function has extreme local curvature properties.

We benchmark PSGD by predicting the Penn TreeBank Dataset using LSTMs using Zhuang et al. (2020)'s framework. Our results (see Table 14 ) indicate that PSGD consistently outperforms (lower is better) other first-order optimization algorithms, including the SoTA AdamW, on 1, 2, and 3 layer LSTMs.

## F.8   POTENTIAL SOCIAL IMPACTS AND LIMITATIONS

Regarding social impact, our optimization method can outperform other optimization methods in generalization accuracy as well as minimizing loss with negligible computational overhead and less tuning efforts due to normalized learning rates. This will enable better training of machine learning systems. Furthermore, since PSGD is better at generalization based on imbalanced datasets, it has the potential to reduce bias and provide better representation for under-represented classes and sub-classes.

The main potential limitation for PSGD we see, is that its certain forms. e.g., low-rank approximation one, require more memory to store the curvature information. Yet, it is still negligible compared to other popular second-order methods like KFAC that require per-sample derivatives.

## F.9   HARDWARE & SOFTWARE

All experiments were run on a single NVIDIA 3080 10GB GPU with an 11th gen Intel i7 processor. We utilized PyTorch Paszke et al. (2017) version 1.13.

For now we have zipped the codebase to re-create our results, and will later host on github.

# G  More on: Lie Groups, Preconditioners and Practical Considerations

As Lie Groups are not a ubiquitous framework for optimization and even less for machine learning, we provide an overview of why we need a general-purpose preconditioner, and practical considerations/timings under different frameworks. Furthermore, we consider different Hessian structures not included in the main paper. We consider fine and coarse grids for future ways to update preconditioners, theoretical connections to PCA, FFT, and DCT preconditioners, and more.

Note we have fundamentals on Lie Groups for low-rank approximations and XMat preconditioners in Appx B & D

## G.1  The Need for a General Purpose Preconditioner

Among the Lie groups listed in Table 15, the Kronecker-product one has been proven successful for many tasks Martens and Grosse (2015); Li (2019); Goldfarb et al. (2020). However, it is inconvenient to use as we need to sort out the parameters to be optimized as a list of tensors in a certain way such that those preconditioners are doing meaningful operations. Here, we are looking for some general-purpose black box preconditioners to avoid the need to rearrange the tensors to be optimized in certain specific ways.

Table 15: Useful Lie groups (one form may have several disconnected groups)

| example forms | parameters | notes |
|---|---|---|
| $\begin{bmatrix} \cdot & & \\ & \cdot & \\ & & \cdot \end{bmatrix}$ | non-zero diagonals | diagonal matrices, scaling |
| $\begin{bmatrix} \cdot & & \\ \cdot & \cdot & \\ & \cdot & \cdot \end{bmatrix}$ | non-zero diagonals | Triangular matrices (lower or upper), feature whitening |
| $\begin{bmatrix} \cdot & & \cdot \\ & \cdot & \cdot \\ & & \cdot \end{bmatrix}$ | non-zero diagonals | feature normalization as in batch normalization |
| $\begin{bmatrix} \cdot & \cdot & \cdot \\ & \cdot & \cdot \\ & & \cdot \end{bmatrix}$ | non-zero diagonals | incomplete triangular matrices |
| $\begin{bmatrix} \cdot & & \cdot \\ & \cdot & \\ \cdot & & \cdot \end{bmatrix}$ | invertible | butterfly matrices, building blocks of Kaleidoscope/FFT/DCT matrices |
| $\begin{bmatrix} \cdot & & \cdot \\ & \cdot & \cdot \\ \cdot & & \cdot \end{bmatrix}$ | invertible | similar to the butterfly matrices, but defined for both odd and even dims |
| $\begin{bmatrix} \cdot & \cdot & \cdot \\ \cdot & \cdot & \cdot \\ \cdot & \cdot & \cdot \end{bmatrix}$ | invertible | plain dense invertible matrices, i.e., the general linear (GL) group |
| $C$ | $C$ is invertible and circulant | cyclic or anti-cyclic group, fast matrix-vector product via FFT |
| $U$ | $U$ is orthogonal/unitary | the groups of rotations (reflections are not continuous) |
| $A$ | $\lvert \det(A) \rvert = 1$ | traceless, $\operatorname{tr}\log(A) = 0$ |
| $C^{-1}AC$ | $A$ is on a Lie group, and $C$ is invertible | useful for blending when $C^{-1}$ is cheap to calculate |
| $U^T A U$ | $A$ is on a Lie group, and $U$ is orthogonal | useful when $U$ is DFT/DCT/Hadamard like transformations |
| $A \oplus B \oplus \ldots$ | $A$ and $B$ are on the same or diff groups | direct sum as in block diagonal matrices |
| $A \otimes B \otimes \ldots$ | $A$ and $B$ are on the same or diff groups | good for matrix/tensor gradient preconditioning |
| $I + UV^T$ | invertible, either fixed $U$ or $V$ | useful for preconditioning via low-rank approximation |
| $\begin{bmatrix} A & B & \ldots \\ C & D & \\ \vdots & & \ddots \end{bmatrix}$ | invertible and all blocks on the same group | large sparse preconditioner construction; special case: butterfly matrix |

## G.2 PRACTICAL CONSIDERATIONS

Clearly, we cannot initialize either $U$ or $V$ or any diagonal element of $B$ to zero. We can only update $U$ and $V$ sequentially. In my implementations, I update either $U$ or $V$ in a single step, determined in a random way, to save computations. I call it the UVd preconditioner. Another form, dUV has the same capacity as shown by

$$(I + UV^T)\text{diag}(d) = \text{diag}(d) + U[\text{diag}(d)V]^T = \text{diag}(d)\left\{I + [\text{diag}(d^{-1})U][\text{diag}(d)V]^T\right\}$$

The Woodbury matrix identity turns $Q^{-T}v$ into solving for a system of $r$ linear equations, where $r$ is the rank of $U$ and $V$. Table 16 summarizes the wall time comparison results on a few typical solvers. We see that their efficiency varies a lot. The fastest one is about two orders of magnitude faster than the slowest one for $r = 10$. A poor combination of hardware and solver could slow down the updating of this preconditioner. Note that theoretically, we could use the Woodbury matrix

Table 16: Mean wall time (ms) over 3000 runs on solving the system of $r$ linear equations, $Ax = b$. Hardware: Xeon (R) W-2135 CPU, and GeForce RTX 2080 GPU.

| | $r = 10$ | $r = 100$ | $r = 1000$ |
|---|---|---|---|
| Matlab (CPU, double, x=A\b) | 0.0078 | 0.10 | 14.2 |
| Numpy (CPU, double, x=np.linalg.solve(A,b)) | 0.17 | 6.7 | 57.2 |
| Scipy (CPU, double, x=scipy.linalg.solve(A,b)) | 0.016 | 0.14 | 20.5 |
| Pytorch (GPU, single, x=torch.linalg.solve(A,b)) | 0.17 | 0.53 | 6.2 |
| Tensorflow (GPU, single, x=tf.linalg.solve(A,b)) | 0.67 | 0.92 | 6.6 |

identity to update the inverse of $I + V^T U$ recursively as well. However, similar to a Kalman filter, this process could be numerically unstable. Directly solving the system of linear equations should be cheap enough for a typical $r$, say $1 \leq r \leq 20$. Again, the low efficiency of certain linear solvers for a small $r$ is another issue, but solvable.

## H HESSIANS WITH CERTAIN STRUCTURES

One import family is the preconditioners for a list of affine transform matrices studied in Li (2019). Here we discuss some other ideas.

### H.1 BAND HESSIAN

The most common assumption is that the Hessian is roughly a band matrix if parameters far away are not strongly coupled. Still, band matrices generally do not form groups, and their inverses are not necessarily band-limited. To obtain a tractable solution, we can approximate $Q$ with

$$Q = (C^{-1}AC)B$$

where both $A$ and $B$ are block-diagonal matrices with block size $K \times K$, and $C$ is a left or right circular shifting matrix that shifts $K/2$ positions. The following equation shows this clearly when $K = 2$ and the first block of $A$ is diagonal.

For preconditioner estimation, we do not need to constrain the two $0.5K \times 0.5K$ anti-diagonal blocks of $A$'s first $K \times K$ block to be zeros (the resultant $Q$ is 'circular' band). Note that the circular shifting matrix is unitary, i.e., $C^{-1} = C^T$. Thus, $P = Q^T Q = (ACB)^T(ACB)$. Hence, we can simply redefine $Q$ as

$$Q = ACB$$

### H.1.1 GRADIENTS

Let us have

$$
\begin{aligned}
dQ &= dACB + ACdB \\
&= \mathcal{E}_1 ACB + ACB\mathcal{E}_2 \\
&= \mathcal{E}_1 Q + Q\mathcal{E}_2
\end{aligned}
$$

Now, it is clear that Eq. equation 19 still can be used to calculate the gradients w.r.t. $A$ and $B$.

Let $dB = B\mathcal{E}$. Then from Eq. equation 19, the gradient w.r.t. $B$ is

$$
0.5\nabla_B = \text{blkdiag}[(Ph)h^T - v(P^{-1}v)^T]
$$

and thus we update $B$ as

$$
B \leftarrow B - \mu B\nabla_B
$$

Similarly, let $dA = \mathcal{E}A$, we can show the gradient w.r.t. $A$ to be

$$
0.5\nabla_A = \text{blkdiag}\{[(Qh)(Qh)^T - (Q^{-T}v)(Q^{-T}v)^T]\}
$$

Then, we update $A$ by

$$
A \leftarrow A - \mu\nabla_A A
$$

As usual, the step size should be small enough such that $\mu\|\nabla_B\| < 1$ and $\mu\|\nabla_A\| < 1$. It is also possible to use block-wise step sizes.

### H.1.2 ON THE ESTIMATION OF $\|\nabla_B\|$ AND $\|\nabla_A\|$

First, the norm of a block diagonal matrix is the maximum norm of its blocks. Second, note that each block has form $ab^T - uv^T$, which has at most rank 2. Thus, $\|ab^T - uv^T\|_F/\sqrt{2} \leq \|ab^T - uv^T\| \leq \|ab^T - uv^T\|_F$. Hence, the maximum Frobenius norm of blocks gives a tight enough spectral norm estimation for the two block diagonal matrices $\|\nabla_B\|$ and $\|\nabla_A\|$.

### H.2 THE TWO-GRID PRECONDITIONER

Inspired by the algebraic multigrid method (AMG), we may precondition on two half-overlapped coarse grids, i.e.,

$$
Q = C^{-1}(A \otimes I)C(B \otimes I)
$$

where $A$ and $B$ are two small matrices, and $C$ is a circular-shifting matrix. The 'coarseness' is determined by the size of $I$. Clearly, it reduces to a dense preconditioner for $I = 1$. When the size of $I$ is large, we only precondition the 'low-frequency components' of the Hessian.

The popular Kronecker product preconditioner also can be viewed as preconditioning on a coarse grid. Unlike AMG, we do not have a prolongation/interpolation step to refine the coarse error since the target is the unknown Hessian.

### H.3 PRECONDITIONING AS PCA OR KARHUNEN–LOEVE TRANSFORM

### H.3.1 THE CONNECTION TO PCA

If $v$ is drawn from $\mathcal{N}(0, I)$, then we can simplify Eq. equation 3 as below

$$
E[h^T Ph + v^T P^{-1}v] = E[h^T Ph] + E\{\text{trace}[P^{-1}vv^T]\} = E[h^T Ph] + \text{trace}(P^{-1})
$$

Then, the optimal $P$ simply whitens $h$ as shown by $E[(Ph)(Ph)^T] = I$. Thus, the optimal preconditioner is performing PCA (principal component analysis). We can even reformulate the preconditioner estimation problem as an online PCA one with a cost

$$
E[\|Q^T Qh\|^2] - 4\log|\det Q| \tag{21}
$$

However, fitting $Q$ in this way converges slowly as this criterion does not exploit all the information encoded in pair $(v, h)$. Still, this connection suggests that we could use certain PCA tools for preconditioning.

### H.3.2 KALEIDOSCOPE, FFT AND DCT MATRICES

If the Hessian has certain sparsities, a Kaleidoscope matrix like $Q$ could do a good job, e.g.

$$Q = \begin{bmatrix} \cdot & & & \cdot \\ & \cdot & \cdot & \\ & \cdot & \cdot & \\ \cdot & & & \cdot \end{bmatrix} \times \begin{bmatrix} \cdot & \cdot & & \\ & \cdot & \cdot & \\ & & \cdot & \cdot \\ & & \cdot & \cdot \end{bmatrix} \times \begin{bmatrix} \cdot & \cdot & & \\ & \cdot & \cdot & \\ & & \cdot & \cdot \\ & & \cdot & \cdot \end{bmatrix}$$

for a $7 \times 7$ Hessian. Theoretically, such a preconditioner has complexity $\mathcal{O}(N \log_2 N)$ for an $N \times N$ Hessian. But, practically, this will take a lot of effort to achieve such efficiency. For now, I rely on the FFT libs to approximate the KLT.

### H.3.3 DCT PRECONDITIONER

Practically, DCT is asymptotically equivalent to KLT for a first-order Markov process with strong adjacent sample correlation. If this is the case, a diagonal or very narrow band preconditioner is sufficient. If the Hessian, $H$, has certain sparsity or regularity like nature signals, e.g., images, then $UHU^T$ will be a highly sparse matrix with most energies concentrated on the upper left corner, where $U$ is the DCT matrix. Hence, we could try a preconditioner like

$$Q = AUB$$

where $A$ is a proper sparse matrix, and $B$ is a diagonal matrix for preconditioning the Hessian-vector products. We call it a DCT preconditioner as $UHU^T$ performs a 2D discrete cosine transform of $H$. Since we are to fit $A$ and $B$ on Lie groups, thus let

$$\begin{aligned} dQ =& dAUB + AUdB \\ =& \mathcal{E}_1 AUB + AUB\mathcal{E}_2 \\ =& \mathcal{E}_1 Q + Q\mathcal{E}_2 \end{aligned}$$

Now, it is clear that we can use the same equation in equation 19 for gradient derivation.

### H.4 PRACTICAL CONSIDERATIONS AND LIMITATIONS

To facilitate the implementations, we may require $N$ to have special values, e.g., $\text{mod}(N, K) = 0$ with block size $K$, or $N = 2^a 3^b 5^c 7^d$ for most FFT libs. If $N$ is not such a number, we could augment the cost as

$$\mathcal{L}(\theta) + 0.5\vartheta^T \vartheta$$

and optimize vector $[\theta, \vartheta]$, which has a conformable length. This trick works well in practice.

All the preconditioners here have certain limitations. Without knowing the block structure, a band preconditioner scales poorly. The DCT preconditioner indeed can de-correlate the input features very well, and thus is good for layer-wise preconditioning, but not necessarily good for preconditioning the whole flattened gradient. Similar to the AMG method, it is very difficult to define a reasonable coarse grid for preconditioning without knowing the connection of weights in the network.

## I PRECONDITIONER ON A SINGLE LIE GROUP

### I.1 DIAGONAL/JACOBI PRECONDITIONER

Possibly the simplest preconditioner with closed-form solution $p = \sqrt{E[v \odot v]/E[h \odot h]}$, where $P = \text{diag}(p)$. A Jacobi preconditioner is not very competitive. Still, it is simple enough for performance study. We have compared three ways for the estimation of $P$: closed-form solution where expectations are replaced with sample averages; updating with a single-step size as

$$p \leftarrow p - \mu(h^2 \odot p - v^2 \oslash p) \odot p$$

where $\mu$ is small enough such that $\mu \max |h^2 \odot p - v^2 \oslash p| < 1$; and updating with element-wise step size as (reduce to sign SGD)

$$p \leftarrow p - \mu \odot \text{sign}(h^2 \odot p - v^2 \oslash p) \odot p$$

where $0 < \mu < 1$. The closed-form solution performs the best, and the single-step size updating may perform slightly better than the element-wise one.

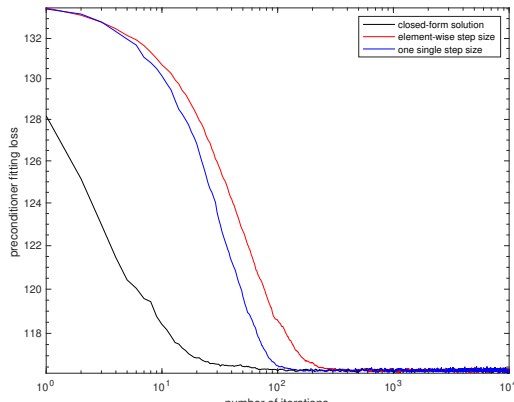

Figure 16: Comparison of three diagonal preconditioner fitting methods on a random dense $100 \times 100$ Hessian with eigenvalues drawn from the standard uniform distribution.

### I.2 X-MATRIX PRECONDITIONER

This simple preconditioner brings a short-cut connection among gradients far away in positions and performs better than the diagonal one. The X-shape can be nested to knit a fishnet-like preconditioner. The butterfly matrices also work well. These simple preconditioners are very light and can be readily adopted to boost the performance of SGD with marginal overhead. Note that these preconditioners can be reduced to the direct sum of smaller groups, i.e., they are reducible.

$$
\text{Diagonal/Jacobi :} \quad
\begin{bmatrix}
\cdot & \\
 & \cdot
\end{bmatrix}
$$

$$
\text{X shape matrix :} \quad
\begin{bmatrix}
\cdot & & \cdot \\
 & \cdot & \\
\cdot & & \cdot
\end{bmatrix}, \quad
\begin{bmatrix}
\cdot & & & \cdot \\
 & \cdot & \cdot & \\
 & \cdot & \cdot & \\
\cdot & & & \cdot
\end{bmatrix}
$$

$$
\text{Fishnet like matrix :} \quad
\begin{bmatrix}
\cdot & & & & \cdot \\
 & \cdot & & \cdot & \\
 & & \cdot & & \\
 & \cdot & & \cdot & \\
\cdot & & & & \cdot
\end{bmatrix}, \quad
\begin{bmatrix}
\cdot & & & & & \cdot \\
 & \cdot & & & \cdot & \\
 & & \cdot & \cdot & & \\
 & & \cdot & \cdot & & \\
 & \cdot & & & \cdot & \\
\cdot & & & & & \cdot
\end{bmatrix}
$$

### I.3 TRIANGULAR MATRIX PRECONDITIONER

Previously proposed in Li (2015) called a dense preconditioner. This calculated the full rank preconditioner and is only applicable to small-scaled problems due to memory and complexity constraints.

## J THE GROUP OF X-MATRIX

All invertible matrices with form

$$A = \text{diag}(a) + \text{adiag}(b)$$

form a Lie group, where $\text{adiag}$ means skew- or anti-diagonal. Clearly, $A$ can be reduced to the direct sum of $\lceil N/2 \rceil$ smaller groups. We assume that the central element of $b$ is zero for $A$ with odd dimensions.

Short-hands:

$ab$ denotes the element-wise product of two vectors $a$ and $b$.

$\overleftarrow{a}$ denotes the flipped version of $a$.

$\text{Proj}_X(A)$ denotes projecting a dense matrix $A$ onto an X-matrix.

Then, we have properties:

$$
\begin{aligned}
[\operatorname{diag}(a) + \operatorname{adiag}(b)]x &= ax + b\overleftarrow{x} \\
\operatorname{diag}(a)\operatorname{diag}(b) &= \operatorname{diag}(ab) \\
\operatorname{diag}(a)\operatorname{adiag}(b) &= \operatorname{adiag}(ab) \\
\operatorname{adiag}(a)\operatorname{diag}(b) &= \operatorname{adiag}(a\overleftarrow{b}) \\
\operatorname{adiag}(a)\operatorname{adiag}(b) &= \operatorname{diag}(a\overleftarrow{b}) \\
[\operatorname{diag}(a) + \operatorname{adiag}(b)][\operatorname{diag}(u) + \operatorname{adiag}(v)] &= \operatorname{diag}(au + b\overleftarrow{v}) + \operatorname{adiag}(av + b\overleftarrow{u}) \\
[\operatorname{diag}(a) + \operatorname{adiag}(b)]^{-1} &= \operatorname{diag}\left(\frac{\overleftarrow{a}}{a\overleftarrow{a} - b\overleftarrow{b}}\right) - \operatorname{adiag}\left(\frac{b}{a\overleftarrow{a} - b\overleftarrow{b}}\right) \\
[\operatorname{diag}(a) + \operatorname{adiag}(b)]^{T} &= \operatorname{diag}(a) + \operatorname{adiag}(\overleftarrow{b}) \\
\|\operatorname{diag}(a) + \operatorname{adiag}(b)\| &\leq \|\operatorname{diag}(a)\| + \|\operatorname{adiag}(b)\| = \max|a| + \max|b| \\
\|\operatorname{diag}(a) + \operatorname{adiag}(b)\| &\geq \max(\max|a|, \max|b|) \qquad (\text{for even dim}) \\
\operatorname{Proj}_X(ab^T) &= \operatorname{diag}(ab) + \operatorname{adiag}(a\overleftarrow{b}) \qquad (\text{for even dim})
\end{aligned}
$$

where for odd dimensionalities, the central element of $\operatorname{adiag}(\cdot)$ must be set to zero so that the last two equations hold as well.

