# OpenReview forum: "Curvature-Informed SGD via General Purpose Lie-Group Preconditioners"
_ICLR.cc/2024/Conference — Submitted to ICLR 2024_

### Official Review · Reviewer_ApxA · 2023-10-30

**Soundness:** 3 good
**Presentation:** 2 fair
**Contribution:** 2 fair
**Rating:** 3
**Confidence:** 4

**Summary:**

In this paper, the authors introduced a general purpose Lie group pre-conditioners for SGD optimization algorithms with application in deep learning problems. They gave som theoretical convergence rate guarantees and empirical showed that the algorithms they proposed outperformed some usual methods in deep learning problems.

**Strengths:**

This submission is well-organized with clear language and structures. The authors gave detailed description and some theoretical analysis for the proposed algorithms. They also conduct a lot of numerical experiments on deep learning problems and these empirical results are pretty good compared with some state-of-art optimization methods. The proposed PSGD seems promising when it is applied to the large scale deep learning problems.

**Weaknesses:**

However, it seems that this paper has the following weaknesses and problems.

1. First question is that in Proposition 2.1, the authors refer to equation 6. But there is no equation 6 in the main part of the paper. Does the authors refer to the equation 6 in the appendix? Then the authors should put this equation in the main part of the submission.

2. In Corollary 2.1.1 the authors mention the linear convergence rate, but what is the exact or explicit upper bound for this linear convergence rate? The authors should explicitly present this linear convergence rate not just present an asymptotical convergence result.

3. In Corollary 2.1.2, the authors claimed that the algorithm can cover the quadratical convergence rate of Newton's method. However, the inequality in this corollary only shows the linear convergence. Because $\|g_t\| \geq 2\alpha(L(\theta_t) - L(\theta_*))$, we can get that $L(\theta_{t + 1}) - L(\theta_*) \leq (1 - \frac{\alpha^2}{\beta^2})(L(\theta_t) - L(\theta_*))$. However, this is only linear convergence rate. It's not quadratical convergence rate as Newton's method. And the linear convergence rate is even worse than the gradient descent of $1 - \frac{\alpha}{\beta}$

4. There is no overall theorem for the convergence rate of the proposed algorithm. We need a specific detailed theoretical convergence rate of the PSGD to make comparison with other methods.

5. The authors didn't give any theoretical analysis for the step size. They only presented some empirical results for the step size choices.

**Questions:**

Please check the weakness.

---

> ### Author Response · Authors · 2023-11-13
>
> Thanks for this reviewer's acknowledgement that the proposed method is promising and performs well in large scale problems.
>
> 1, On equation 6.
>
> Yes, you can click it to bring you to the eq. 6 in the appendix. We will try to squeeze that equation into the main body to improve reading experience.
>
> 2, On corollary 2.1.1.
>
> Corollary 2.1.1 is just the immediate results of proposition 2.1 and text book results. We just want to show that PSGD recovers SGD. One can follow the standard technique for SGD to give a specific convergence rate for PSGD for each specific setting. We have no intent to elaborate in this direction.
>
> 3, on corollary 2.1.2
>
> Again, corollary 2.1.2 is just the immediate results of proposition 2.1 and text book results. We just show that PSGD recovers Newton's method. The Newton method does not always have quadratic convergence, it only converges quadratically around the basin of attraction. One can follow the standard textbook technique to decide the size of that basin of attraction. We do not claim any stronger results, no intention to elaborate in this direction.
>
> Still, we apologize that the presentation of corollary 2.1.2 can be improved to avoid such possible misunderstanding. Will revise it.
>
> 4, Specific convergence rate of PSGD.
>
> Again, Proposition 2.1 says that P must converge to inv(abs(Hessian)). Then one can follow the standard techniques to give a specific convergence rate for PSGD in each case, by specifying settings like a) Lipschitz constant, 2) learning rate,  3) distance between initial guess to final solution, 4) alpha, 5) beta in corollary 2.1.2, ....
>
> PSGD just provides inv(abs(Hessian)) as a preconditioner, and thus gives you a learning rate free method, and recovery of Newton method for convex problems. We do not intend to prove any stronger results in corollary 2.1.1 and 2.1.2, also not the focus on the paper. Still, corollary 2.1.1 and 2.1.2 are not nonsense as not all preconditioners can recover the Newton method in a convex setting.
>
> 5, step sizes.
>
> All the step sizes are normalized and scaleless in theory, not empirically. It’s completely a learning rate free method. On Lie group, you update as  (I + step * Generator) * Q, and thus you know how much you updated by checking ||step * Generator||. For parameters, all the eigenvalues of P * Hessian have unit absolute value, and again, you know how much you have updated the parameters as ||delta preconditioned_gradient|| = ||delta parameters||. Nevertheless, a scheduler is needed in a stochastic setting, and we do need to tune the scheduler.

---

### Official Review · Reviewer_rA7r · 2023-10-31

**Soundness:** 2 fair
**Presentation:** 2 fair
**Contribution:** 2 fair
**Rating:** 5
**Confidence:** 3

**Summary:**

This paper leverage the symmetry of Lie group constraint to simplify the preconditioner fitting process, which eventually accelerate stochastic gradient descent by utilizing curvature information obtained from Hessian vector products or finite differences of parameters and gradients. Experiments shows that the claimed preconditioned SGD outpuerforms sota in different area of artificial intelligence such as computer vision, NLP and RL. This is an interesting submission, different from previous results in literature, current work shows that the proposed approach works well with naturally normalized parameters instead of using damping or line search.

**Strengths:**

1. The experiments of this paper is very sufficient and clearly strengths the theoretical claims.
2. Despite the idea of leveraging symmetry of Lie group in deep learning has been discovered and studied before, the scenarios  of application of simplifying and accelerating SGD is novel.

**Weaknesses:**

1. From the presentation of section 2.3, the "Lie group" only refers to orthogonal groups and general linear groups, which is a much more limited case than general Lie groups.
2. A brief introduction on Lie groups is needed, at least in appendix, this can be part of section D.1. Since gradient descent on manifold has been mentioned as well, it is necessary to define it formally, especially the form that is used in the Lie groups of this paper.
3. Same issue happens in experiments. There are much more experiments than a theory oriented paper, but it is hard to follow the whole experiment sections due to current presentation.
4. Inconsistence of terminology. In sentences, "Theorem" is used but in presentation of the results formally, "Claims" have been used for no reason.

**Questions:**

1. In each experiment, what is the specific constraints?
2. In proposition 2.1, how is the step-size set?
3. Contribution of theoretical analysis can be stronger, there are sufficently many works on optimization on manifolds, what is the main challenging of this paper compare to other related works? Is that possible to leverage other works including rates to enhence the results of this paper?

---

> ### Author Response · Authors · 2023-11-13
>
> We appreciate this reviewer's acknowledgement that the proposed method is novel and empirical results are sufficient.
>
> 1, Only limited to orthogonal or general linear group?
>
> No. We do not limited to orthogonal group. We only use orthogonal group to illustrate that coordinate change consists of rotation and scaling.
>
> Yes. We assume a general linear group to prove Proposition 2.1.
>
> No. The two proposed preconditioners (matrix-free and low-rank approximation) are not limited to either orthogonal or general linear group. They all are sparse, and have linear complexity (the same order as SGD and Adam). The orthogonal and general linear group have quadratic complexity, which means that they need 10000000 billion elements for preconditioning a network with 100 millions parameters!
>
> 2, Introduction of Lie group. We do have some background materials on Lie groups in appendix D and G. Readers with knowledge of group generators should find them easy to follow; we hope that readers knowing the definition of Lie group can follow them as well.
>
> 3, Much more experimental results than a theory oriented paper.
>
> We do not intend to have a theory oriented paper. The general theory of PSGD is established in previous Li’s work (the ICLR 2018 one). However, that work only considers affine groups and thus have three limitations: a) not all network blocks can be reparameterized as affine transforms, although many, say CNN and attention, can; b) no interaction among affine transform preconditioners of different layers or blocks; c) it is inconvenient to use the affine group preconditiioner as it requires to reparameterize all blocks into their affine transform forms. We fully disclosed these backgrounds in Section I and II. One major contribution of this paper is the low-rank approximation (LRA) preconditioner. The math for LRA preconditioner on Lie group is much heavier than in the Euclidean space. This makes our paper full of math equations, still we do not intend to make it a theory oriented paper.
>
> Readers could just focus on the results listed in the main body. We provide so much more experimental results because even up to today, most people still think that 2nd order optimizers only work for small scale toy problems (although true for BFGS, LM-BFGS, HF, CG, …). We really hope that all these readily reproducible experimental results in this paper could let people give 2nd order optimizers a second look.
>
> 4, Inconsistency of terminology.
>
> Apology for the confusion. We will correct them. Our intentions for these terms are:
> Proposition) important theoretical results;
> Corollary) immediate results from propositions and known results;
> Claims) can be proved with straightforward algebra operations, although still could be math heavy.
>
> 5, Constraints for each experiment.
>
> We guess the reviewer is inquiring about resource constraints. We have details in the appendix, and code for reproduction. A rough idea of the complexity of LRA method (the other one is cheaper): a) 1.2 x wall clock time of SGD; b) 2~3 x memory consumption of SGD; c) less tuning time as all the step sizes are normalized.
>
> 6, Step size in proposition 2.1.
>
> On the Lie group, Q is updated as (I + step * Generator) * Q. To make an infinitesimal update, we need || step * Generator || << 1. Then we can choose the step as a_normalized_step / || Generator ||, where a_normalized_step << 1, say 0.01. This way of normalization does not always work. But here it well works (reason: see algorithm 2 in page 5, the two terms of nabla_U are homogeneous). Also, the spectral norm estimation is tight (within 3 dB, see D.1.2.2). So no implementation difficulties.
>
> 7, Stronger theoretical results? Agree that the theoretical results (corollary 2.1.1 and 2.1.2) are pretty plain, mainly for the completeness of the paper. Our focus is on the practical side: how to construct the LRA preconditioner and how effective it is on large scale ML problems. Difficulties for theoretical studies are from two sides. a) the PSGD preconditioner fitting criterion does not have a closed-from rank-2 preconditioner update rule even in the Euclidean space (BFGS criterion has, but does not work for large scale ML problems as it ignores gradient noises). b) Update on Lie group is nice as we can normalize the step size and have all the nice properties of Lie group. But, Lie group is a manifold plus extra constraints: the way we update the preconditioner cannot break the closeness of the Lie bracket. Still, a good direction worthy closer check.

---

### Official Review · Reviewer_PCFY · 2023-10-31

**Soundness:** 3 good
**Presentation:** 3 good
**Contribution:** 2 fair
**Rating:** 5
**Confidence:** 5

**Summary:**

This paper suggests a quasi-Newton type method with updates of preconditioner made according to minimum of criterion, which takes into account stochasticity of the gradient. Optimization of preconditioner is performed on the subgroups of general linear group. Considering small and expressible enough subgroups leads to algorithms with moderate computational complexity and fast convergence in practical tasks.

**Strengths:**

The methods proposed are indeed practically efficient, and developing the concept of optimizable preconditioners is topical.

**Weaknesses:**

Theoretical frameworks is mostly ihnerited from that in Xi-Lin Li's papers, especially one that is titled "PRECONDITIONER ON MATRIX LIE GROUP FOR SGD" and contains the majority of ideas ensuring success of an approach of the paper under review. The constructions of particular subgroups and detailed reasoning of why operations are correct are of interest, but span of proof has been already described in original Lin's paper.

**Questions:**

1. Can you provide convergence rate guarantees which do not include conditioning number? The presence of condition number in rate makes rate similar to that of SGD without preconditioning. Taking into account, that purpose of preconditioning is in fact getting rid of conditioning number impact, theory seems to fail (in non-convex case).
2. Another criterion from Lin's original papers leads to Fisher matrix as an optimal preconditioner. Can you explain why this important case was left outside of the framework you propose? This case is interesting because it has no quasi-Newton property, but is greedy-optimal for quaite natural criterion at the same time. Whithout considering both cases, your approach seems to be just a detalisation of particular case of Lin's approach

---

> ### Author Response · Authors · 2023-11-13
>
> We appreciate this reviewer's acknowledgement that the proposed method is efficient.
>
> 1, Overlap with Li’s ICLR 2018 work.
>
> Yes, criterion for preconditioner estimation and the idea of Lie group fitting are from Li’s previous work. We fully disclosed these details in Section I and II.
>
> But, the affine Lie group preconditioner developed in Li’s ICLR work have three limitations: a) not all network blocks can be reparameterized as affine transforms, although many, say CNN and attention, can; b) no interaction among affine transform preconditioners of different layers or blocks; c) it is inconvenient to use the affine group preconditiioner as it requires to reparameterize all blocks into their affine transform forms.
>
> So this paper proposed two black box style preconditioners. One can use them as easily as the plain SGD. Notably, the low-rank approximation (LRA) preconditioner is of significant theoretical importance (LRA tends to fit real world patterns well) and practical importance (makes PSGD have the same order of complexity as SGD, equally easy to use).
>
> 2, Convergence rate without condition number?
>
> Such a picture makes intuitive sense, but still many hinders in math. PSGD just gives a learning rate free method as P*H has unit absolute eigenvalues, but loss still could be arbitrarily rough, like a large Lipschitz constant, and could be full of saddle points for nonconvex problems. A convergence rate independent of condition number would be extremely strong! Even for the standard Newton algorithm on strictly convex problems, text book results can only show that the Newton method converges quadratically around a small basins of attraction, generally no globally quadratic convergence. Away from the attractor, Newton method with reduced step size also converges linearly as gradient descent.
>
> To clarify, Proposition 2.1 is completely ours, stating that P must converge to inv(abs(Hessian)). Corollary 2.1.1 and 2.1.2 are the immediate results of Proposition 2.1 and established results from convex optimization. Corollary 2.1.1 says that PSGD is no worse than SGD for nonconvex problems, and Corollary 2.1.2 says that PSGD recovers the Newton method for convex problems. We claim no stronger results. Sounds a little pessimistic in theory, but in practice, a good preconditioner helps a lot for sure as shown in our empirical results. At least, less tuning effort due to normalized learning rate.
>
> 3, Why no Fisher-type preconditioner?
>
> It’s true that we only consider the Newton-type preconditioner. This paper benchmarks PSGD on a wide range of problems, from classic convex optimization to reinforcement learning. The Fisher-type is for sure irrelevant for some of the problems. Nevertheless, we agree with the reviewer that this is a direction worthy of a closer look. Luckily, both types share the same math. The same LRA tool also applies to the Fisher-type.
>
> Why Fisher-type is not good for general optimizations. a) the Fisher-type works only if the Fisher information matrix can be defined. Unfortunately, this is not always the case. Many regression and reinforcement learning problems do not use divergence or xentropy as losses. b) Let’s narrow down to problems with Fisher information matrix defined. Then, certain regularizations, say L2, could be absorbed by adding a damping term to the Fisher matrix. Still, many other regularizations, say norm of gradients, are incompatible with the use of Fisher information matrix. c) Let’s put aside the regularization term. Then we face two choices for the Fisher: empirical Fisher or the true Fisher. Using the empirical Fisher could be problematic. Using the true Fisher makes the Fisher-type method reduce to the Gaussian-Newton method for pdf from the exponential family, which eventually is an approximation of the Newton-type method by discarding the cross-terms in the Hessian.

---

### Official Review · Reviewer_eh9x · 2023-11-01

**Soundness:** 3 good
**Presentation:** 1 poor
**Contribution:** 3 good
**Rating:** 5
**Confidence:** 3

**Summary:**

In this paper authors work on the preconditioning technique for SGD. They propose two novel preconditioning approaches, based on Lie groups. One preconditioner is Sparse Matrix-Free preconditioner, the other one is Low-Rank Approximation preconditioner. These preconditioners allow SGD to converge linearly, if Hessians eigenvalues are well-bounded, and quadratically, if objective is smooth and strongly convex.

**Strengths:**

1. A lot of various numerical experiments
2. Proposed techniques show superiority over other existing approaches in practice.

**Weaknesses:**

1. Unfortunately, since I am not familiar with Lie groups, it is rather hard for me to fully understand the contribution of the work. If you could provide more background on this topic (maybe even in Appendix), it would be very helpful.
2. It is rather weird to see theoretical convergence results in the Background section. If this is not your result, please provide the citation of the work, where from you have taken this result (and other theorems/corollaries). If this is your result, please move it to other section with results.

**Questions:**

Maybe, you could provide any directions for future research?

---

> ### Author Response · Authors · 2023-11-13
>
> We appreciate this reviewer's acknowledgement that the proposed method is effective.
>
> 1, Background on Lie groups or PSGD?
>
> Lie group) We put some background material on Lie groups in appendix D and G. Nevertheless, this is a conference paper, we keep them short. Maybe particle physicists have nice Lie group introduction materials.
>
> PSGD) The general theory of PSGD and PSGD on the affine Lie group are not new (Li’s ICLR 2018 work). We disclosed them fully in Section I and II. But, PSGD on affine Lie group have three limitations: a) not all network blocks can be reparameterized as affine transforms, although many, say CNN and attention, can; b) no interaction among affine transform preconditioners of different layers or blocks; c) it is inconvenient to use the affine group preconditiioner as it requires to reparameterize all blocks into their affine transform forms.
>
> 2, Contributions.
>
> Contribution I) We propose two black box style preconditioners so that one can use them just as easily as torch.optim.SGD. This is why we can benchmark PSGD on such a wide range of problems in this paper. The low-rank approximation (LRA) one is of high theoretical importance and practical importance.
>
> Contribution 2) We showed that the proposed 2nd order optimizers work well on large scale ML problems. Not a small piece as most 2nd order optimizers (BFGS, LM-BFGS, HF, CG, …) are not as competitive as SGD for scale ML problems.
>
> The convergence analysis (proposition 2.1, corollary 2.1.1 and 2.1.2) is not our focus, just for the completeness of the paper. Still, it is a nice contribution.
>
> 3, Putting convergence results in the background?
>
> Proposition 2 is completely ours. Corollary 2.1.1 and 2.1.2 are the immediate results of Proposition 2 and established results from optimization theory. We will move them into Section 3.
>
> 4, Direction of future research. We vaguely feel that the LRA Hessian estimator could open a door to many possibilities. One thing interesting is to use it for the understanding of the generalization ability of models from the view of information theory.

---

> > ### Comment · Reviewer_eh9x · 2023-11-20
> > **Enhancement of the presentation**
> >
> > Thanks to authors for the comments. But still it seems to me, that presentation of the idea is very bad and needs to be significantly enhanced. For example, you could introduce a separate section with motivation. Now it seems like it is mixed with some minor theoretical results and background and notations in section 2, which is very confusing.
> > I understand that the volume of conference paper is very limited, but you could shrink the experimental part somehow or move some of experiments to appendix. To my opinion the main purpose of any paper is to present your new idea, which includes but not limited to showing better theoretical and/or experimental results.

---

> > > ### Comment · Reviewer_eh9x · 2023-11-22
> > > **Enhancement of the presentation (2)**
> > >
> > > Sorry, but mentioned problems seem crucial to me, so I decrease my rating of the paper.

---

> ### Author Response · Authors · 2023-11-22
> **Motivation**
>
> Please see Introduction and Background for motivation details summarized here.
>
> Motivation:
> 1) not all network blocks can be reparameterized as affine transforms, although many, say CNN and attention, can;
> 2) no interaction among affine transform preconditioners of different layers or blocks;
> 3) it is inconvenient to use the affine group preconditiioner as it requires to reparameterize all blocks into their affine transform forms.
> 4) have a general purpose second-order method that does not require dampening and line search and is robust across hyper-parameters.
>
> So this paper proposed two black box style preconditioners. One can use them as easily as the plain SGD. Notably, the low-rank approximation (LRA) preconditioner is of significant theoretical importance (LRA tends to fit real world patterns well) and practical importance (makes PSGD have the same order of complexity as SGD, equally easy to use).  We certainly state these in the Introduction.
>
> On the reviewers comment. "To my opinion the main purpose of any paper is to present your new idea, which includes but not limited to showing better theoretical and/or experimental results."
>
> We make claims, for example we are robust to gradient noise (in intro), explain them theoretically (in background), give an easy to understand toy experiment (5.1 Fig 1.b), and then do a more complicated label noise experiment (Fig2 c) and show we outperform many SOTA optimziers, and put this in the main of the paper. But if one looks to Appendix F.1 (that we refer to at end of 5.2) we also consider other label noise experiments as well as more optimziers such as SAM. We also have gradient noise in the convex setting where we add Bernoulli noise to the training images see F.5 & Fig. 11.  We make sure to backup our claims with experiments and we in fact put many of them in the Appendix.
>
> At the end of the day, PSGD benefits from preconditioning on the Lie Group and the preconditioner styles. Unfortunately, Lie Groups are not well understood in the ML community, this paper provides a strong LRA to fit the curvature information and a light weight XMat approximation as well. We show these versions of PSGD work very well in practice.

---

> ### Comment · Reviewer_eh9x · 2023-11-22
>
> Thanks for your response. But seems like you did not understand my point. Let me clarify. I am not telling that your result is bad or insignificant. To my opinion, the **presentation** of your idea needs some serious polishing.
> 1. First of all, you haven't fixed my remark about moving your minor propositions and claims from the background section. Even if you think they are insignificant, it is misleading, as I mentioned in my review.
> 2. To my opinion, Introduction part should introduce person to the problem you are solving and the methods, that you are using. In your paper it is a mix of Introduction and Literature review. Usually readers do not want to dive in Literature review before understanding the motivation, main idea and results. You deprive readers from such ability, dumping on them a lot of background information.
> 3. Section *Notations* obviously has wrong name, because most of it consists of problem formulation, that need to be moved to *Introduction*, and a bit of motivation. Thus, this section should be split somehow and its contents moved to other sections.
> 4. In section *The preconditioning fitting criterion* you continue describing used approach, that you started to describe in *Notations* section. It should be in *Introduction* or as a part of *Main results* section
> 5. Section *Preconditioners on Lie Groups* again describes your approach and provides some motivation in the last paragraph.
> 6. Maybe you'd better even to move all the motivation to different subsection after *Introduction*.
>
> Overall, the first two sections, that should **effortlessly** introduce person to solved problem, present the motivation and used methods, represent a mess of badly structured text, that needs to be read several times to understand, what authors are trying to present. Instead of highlighting the proposed approach (not results, that this approach achieves, I emphasize), authors (most likely unintentionally) hide it inside lots of text and background information.
>
> 7. Finally, when I read the section *General Purpose Lie Group Preconditioners*, I had no idea that this is a presentation of your approach. Again, you hide it instead of trying to present.

---

> > ### Author Response · Authors · 2023-11-23
> > **Response to Reviewe's comments.**
> >
> > 1. First of all, you haven't fixed my remark about moving your minor propositions and claims from the background section. Even if you think they are insignificant, it is misleading, as I mentioned in my review.
> >
> > The paper focuses on LRA and Xmat preconditioners. The information in the background is in fact background information and properties that we are bringing to light for the remainder of the paper. This sets the background for the paper. The claims, in section 3 are novel claims (regardless of how simple they may or may not be) that are independent to previous work on PSGD and are not background. Moving it does not make sense.
> >
> > 2. To my opinion, Introduction part should introduce person to the problem you are solving and the methods, that you are using. In your paper it is a mix of Introduction and Literature review. Usually readers do not want to dive in Literature review before understanding the motivation, main idea and results. You deprive readers from such ability, dumping on them a lot of background information.
> >
> > We strongly disagree with your opinion. We feel an intro section should talk about the leading and relevant methods in the literature to frame the problems we will try to solve. As we mentioned before, we talk about the motivation in the intro section. One can find relevant background in section 2 entitled Background.
> >
> > 3. Section Notations obviously has wrong name, because most of it consists of problem formulation, that need to be moved to Introduction, and a bit of motivation. Thus, this section should be split somehow and its contents moved to other sections.
> >
> > Putting formulations in an intro section is unorthodox.
> >
> > 4. In section The preconditioning fitting criterion you continue describing used approach, that you started to describe in Notations section. It should be in Introduction or as a part of Main results section.
> >
> > We again kindly disagree. Putting theory in the intro does not make sense. The Preconditioning Fitting Criterion in fact falls under 'background,' as "We adopt the preconditioner fitting criterion proposed in Li (2015)." Which is the first line of that section. This is neither intro nor a main result.
> >
> > 5. Section Preconditioners on Lie Groups again describes your approach and provides some motivation in the last paragraph.
> >
> > Yes, that section does describe the general background that are needed for PSGD as a optimization framework. At the end of the section entitled 'Preconditioners on Lie Groups', we have two transition sentences, that serves to bridge the background to the main results of the paper, re-affirming the motivation.
> >
> > 7. Finally, when I read the section General Purpose Lie Group Preconditioners, I had no idea that this is a presentation of your approach. Again, you hide it instead of trying to present.
> >
> > At the end of section 2 we say "in this paper, we propose two types of novel general purpose preconditioners." And then immediately after we have a section named "General Purpose Lie Grounp Preconditioners," that we propose two types of general purpose preconditioners, "Matrix-Free Preconditioners" and "Low-Rank Preconditioners." These are clearly stated in our abstract. Nothing was hidden as the reviewer states.
> >
> > We kindly thank the reviewer for their time and comments.

---

### Meta-Review · Area_Chair_4DuY · 2023-12-06

**Metareview:**

The reviewers are unanimous that the paper contains encouraging experimental results and an interesting and important message. However, there are concerns on the presentation of the material. The authors acknowledge that the theoretical aspects overlap with a prior work of Li's in ICLR 2018, and that the paper's contributions are primarily experimental. However when viewed in this manner, the paper is unfortunately written in a way that makes it less accessible for a practitioner with only basic familiarity with Lie groups, who might only be looking to apply the techniques in this paper for a practical problem. As such, I am in agreement with the reviewers that the paper is not yet ready for publication.

**Justification For Why Not Higher Score:**

The weaknesses outweigh the strengths

**Justification For Why Not Lower Score:**

N/A

---

### Decision · Program_Chairs · 2024-01-16

Reject